

# Unprecedented Twenty-First Century Glacier Loss on Mt. Hood, Oregon, U.S.A.

Nicolas Bakken-French[1], Stephen J. Boyer[1], W. Clay Southworth[1], Megan Thayne[1], Dylan H. Rood[2]*, and Anders E. Carlson[1]

[1]Oregon Glaciers Institute, Corvallis, OR 97330, U.S.A.
[2]Department of Earth Science and Engineering, Imperial College London, London SW7 2AZ, U.K.

*Correspondence to*: Dylan H. Rood (d.rood@imperial.ac.uk)

**Abstract.** As part of the southern Cascades, Mt. Hood is the tallest and most glaciated peak in Oregon, U.S.A. Despite alpine glaciers being one the clearest indicators of human-caused climate change, the 21st century behavior of glaciers on

Mt. Hood has not been directly documented. Here we directly measure changes in Mt. Hood's glacier extents from 2003 to 2023 and find dramatic retreat of all glaciers, with one glacier disappearing, another two nearing this status, and a third retreating towards this status. The seven largest glaciers on the volcano lost ~2.8 km$^2$, or ~40% of their area in the 21st century. Comparison to historic records of glacier area back to 1907 shows that this 21st-century retreat is unprecedented with respect to the previous century and has outpaced modeled glacier changes. The rate of retreat over the last 23 years is

more than double the fastest rate documented in the last century from 1907 to 1946. We demonstrate that this century-scale retreat strongly correlates with regional 30-year-average climate warming of ~1.1ºC since the early 1900s, but not with regional changes in precipitation. We conclude that Mt. Hood's glaciers are retreating in response to a warming climate and that this recession has accelerated in the 21st century, with attendant consequences for water resources.





## 1 Introduction

The global retreat of glaciers in the 21st century is accelerating (Zemp et al., 2015; Hugonnet et al., 2021) with century-scale glacier recession attributed to regional climate change in response to anthropogenic greenhouse gas emissions (Marzeion et al., 2014; Roe et al., 2017, 2021). The 2021-2022 hydrological year was the 35th consecutive year of negative annual balance for 50 reference glaciers on six continents (Pelto, 2023). If global average temperature increases by 1.5°C relative to the pre-industrial period, the limit set forth in the Paris Agreement, roughly half of the world's individual glaciers are

projected to disappear (Rounce et al., 2023). Indeed, in the western United States, this deglaciation is already coming to fruition. The Trinity Alps in the State of California became a glacier-free mountain range in the last decade (Garwood et al., 2020). The Olympic Mountains of Washington State have lost 35 glaciers since 1980 (Fountain et al., 2022). Eleven of the 37 named glaciers in Glacier National Park in the State of Montana ceased to be classified as glaciers as of 2015 (Fagre et al., 2017). Washington State's Mt. Rainier, the most glaciated peak in the contiguous United States, has had one glacier

recently disappear with two more nearing stagnation (Beason et al., 2023).

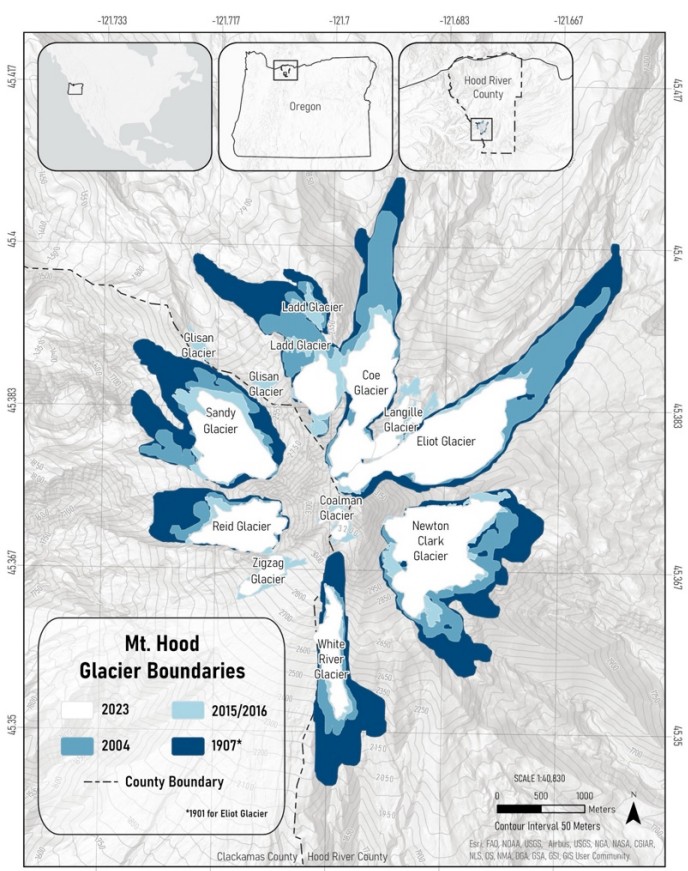

**Figure 1: Location map of United States, Oregon, Hood River County and Mt. Hood. 1907/1901 and 2004 ice extents from Jackson & Fountain (2007); 2015/2016 ice extents from Fountain et al. (2023). 2023 extents from this study. Contour interval is 50 m.**



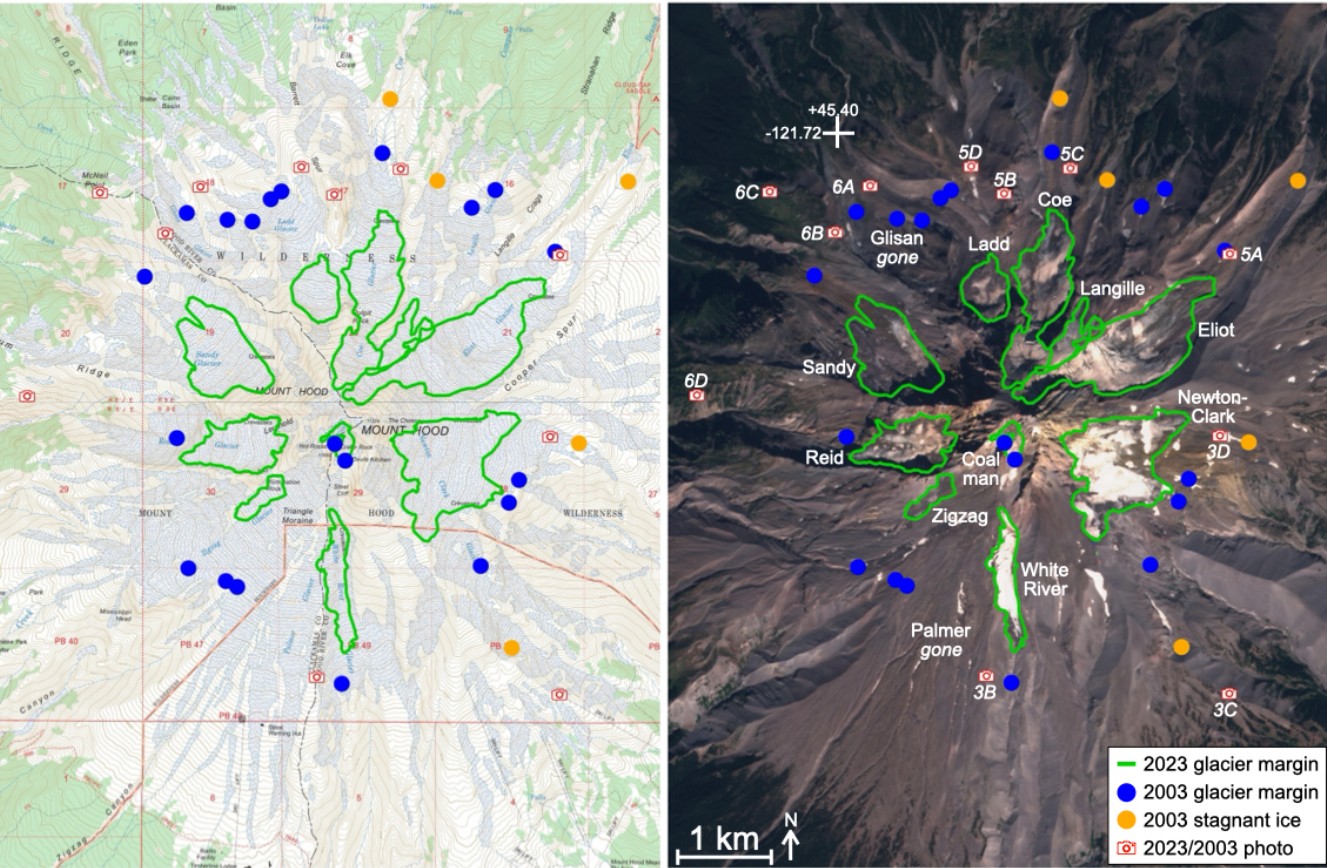

**Figure 2: U.S. Forest Service 2016 topographic map of Mt. Hood (left) and 19 September 2023 satellite image of the same region (right). Green lines are September 2023 outlines of remaining active glaciers. Blue symbols are the lowest termini of glaciers in September 2003 while orange symbols are the lowest elevations of stagnant ice in September 2003. Red camera symbols denote repeat photography locations with Fig. 3, 5 and 6 identifications noted on right.**

Despite these ominous predictions and dire observations for glaciers, data on the recent behavior of glaciers in the Cascade Range of Oregon are limited (Fig. 1). Glaciers in Oregon were first noted in 1870 on Mt. Hood, the tallest and most glaciated peak in Oregon, only days after the first glaciers in the contiguous United States were recorded on Mt. Shasta in California (O'Connor, 2013). Changes in Mt. Hood's glaciers were monitored by the Research Committee of the Mazama Mountaineering Club from the 1930s to mid-1980s (e.g., Phillips, 1935, 1938; Mason, 1954; Handewith Jr., 1959; Dodge, 1971, 1987). However, the extent of Oregon Cascade glaciers on current U.S. Geological Survey and Forest Service maps are still based on air photos from the 1950s (e.g., Fig. 2) (Fountain et al., 2017). The glaciated area of Mt. Hood was last measured with field observations in 2004 (Jackson and Fountain, 2007) whereas glacier lengths with field observations were last recorded in 2000 (Lillquist and Walker, 2006). Fountain et al. (2023) documented snow and ice areas in the western United States using 2015 and 2016 imagery for Mt. Hood's glaciers but did not test mapped accuracy with field observations. The timing of this imagery also misses what multi-year and longer impacts the 2015 record-low snowpack



(Mote et al., 2016) and the June 2021 Pacific Northwest heatwave (Philip et al., 2022; Thompson et al., 2022) would have on Mt. Hood's glaciers (e.g., Pelto, 2018; Pelto et al., 2022).

In terms of climate change impacting Oregon's glaciers, Lillquist and Walker (2006) did not find a relationship between glacier-length change on Mt. Hood and temperature or precipitation change over the 20th century. Similarly, Menounos et al. (2019) documented minimal elevation change for Mt. Hood's glaciers from 2000 to 2018, interpreting from this an
approximately neutral mass balance for the first 19 years of the 21st century. A near zero mass balance for 2000-2018 is at odds with the mass changes for global reference glaciers that include four glaciers to the north of Mt. Hood in the state of Washington; these four all had cumulative negative mass balances for the same time period (Pelto, 2023). Climate-glacier modeling also simulates the emergence of glacier retreat from natural variability in western North America since the 1960s (Marzeion et al., 2014). As such, recent impacts of human-caused climate change on Mt. Hood's glaciers have been unclear
and largely unexplored from direct field observations.

In addition to deciphering the regional impacts of human-caused climate change, glacier change on Mt. Hood is of import to downstream ecosystems and economies. Orchards and farms in the Hood River Valley depend on Mt. Hood's glacier meltwater, particularly in late summer when glacier meltwater can supply up to ~70% of streamflow (Nolin et al., 2010; U.S. Bureau of Reclamation, 2015; Frans et al., 2016, 2018). Unique microbiomes recently described on Mt. Hood depend on
glacier-fed spring water (Miller et al., 2021). Mt. Hood glacier meltwater also maintains cold instream temperatures and late summer streamflow upon which salmon rely (Pitman et al., 2020; Thieman et al., 2021). As such, future water resource management plans for the Hood River basin included projections of future glacier change on Mt. Hood (U.S. Bureau of Reclamation, 2015; Thieman et al., 2021). In addition, Mt. Hood is one of the most climbed mountaineering objectives in the world (~10,000 attempts per year), which is becoming more dangerous during a diminishing climbing season with greater
rockfall as the volcano's cryosphere has changed in recent years (O'Neil, 2023).

Here we document glacier change on Mt. Hood over the last 20 years. We conducted repeat photography and field mapping of glacier extents in 2003 and 2023, which combined with satellite imagery determine the magnitude of glacier change up to mid-September 2023. We place these 21st-century glacier changes in the context of 20th-century glacier changes and test the relationship between ~120 years of glacier-length variability and regional climate change.

## 2 Methods


In 2003 (28 August to 4 October), the lowest actively flowing terminus positions and stagnant-ice positions were measured (latitude-longitude-altitude) by global positioning system (GPS) for every glacier on Mt. Hood (Fig. 2); repeat measurements and comparison to known elevation markers determined an elevation uncertainty of ±6 m. Termini were photographed with locations noted for future repeat photography (i.e., 2023). Figure 2 shows the location of the repeat photographs. Mt. Hood is



an active volcano, with high amounts of rock fall and variations in geothermal heat and topographic relief that influence accumulation, ablation, ice flow, and ice extent (e.g., Lundstrom et al., 1993; Lillquist and Walker, 2006; Jackson and Fountain, 2007; Howcutt et al., 2023). For this reason, all glaciers were described in detail in the field.

From 15 to 22 September 2023, repeat photographs were taken at the same locations as the 2003 photographs and new comparable field observations were made for all of the glaciers. We selected photographs to repeat from an archive of
photographs taken in 2003. Using the image and field notes regarding the location of the images from 2003, we located the precise site of each selected 2003 photograph. The same frame was maintained using relatively unchanged features such as buttresses, cliffs, and moraines as points of reference.

For the 2023 extent of glaciers on Mt. Hood, we initially mapped ice extent using composite Google Earth imagery. As these span multiple years and are not necessarily recent, we then refined ice margins using weekly Sentinel satellite images that
were taken just before the first snowfall. These outlines were refined over four years for the summers of 2020, 2021, 2022, and 2023, which allowed for precise determination of margins. In October of 2020, September of 2021, and September of 2023, we mapped glacier extents in the field and checked these glacier limits for all the glaciers on Mt. Hood. Debris-covered termini are common on Mt. Hood glaciers, and such field mapping of the termini is critical to accurately delineate glacier margins (e.g., Lundstrom et al., 1993; Ellinger, 2010). Specifically, glacier margins were walked with GPS and
debris-covered stagnant ice was separated from debris-covered active ice. In all cases, we found excellent agreement between our remote-sensed ice limits and the field-observed ice limits. As such, any uncertainty in these field-based glacier extents is difficult to quantify as the values are small. Using only remote sensing, Fountain et al. (2023) had their lowest ice-extent uncertainty for a Mt. Hood glacier at ±1%, while Jackson and Fountain (2007) reported an uncertainty for their 2000 Mt. Hood glacier areas of ±0%. Our uncertainty is most likely lower than ±1% but to be conservative we assume a ±7%
uncertainty in the glacier area for 2023, which is the average area uncertainty for the 2004 extents of Mt. Hood glaciers (Jackson and Fountain, 2007).

To place the rate of glacier area loss on Mt. Hood for the 21st century in the context of the 20th century, we compare our 2023-measured area with the glacier area records of Jackson and Fountain (2007) for the time period 1907 (or 1901 in the case of Eliot Glacier) to 2004 for seven glaciers on Mt. Hood: White River, Newton-Clark, Eliot, Coe, Ladd, Sandy, and
Reid. Because of the different temporal resolutions of each glacier-area record, we focus on area change between observational years common to all seven glaciers: 1907, 1946, 1972, 2000, and 2023. The one exception is Newton-Clark Glacier, which lacks an aerial extent for 1946. We extrapolate between the measured extent in 1935 and 1956 when the glacier lost ~0.04 km$^2$ over these 21 years. We focus on the change from 2000 to 2023, rather than 2000 to 2004 and then 2004 to 2023, to make the duration over which the most recent period of change is measured closer to the earlier temporal
spacing. In calculating the rate of change, we propagate through the uncertainty in the area from Jackson and Fountain (2007) and our own 2023 areas. In the case of the 2000 area, we conservatively apply a ±7% uncertainty.



In the early 1900s, the conical shape of Mt. Hood allowed some glaciers to expand laterally at lower elevations, giving them broad ablation areas relative to narrower accumulation areas (Fig. 1) (Jackson and Fountain, 2007). This could make initial glacier-area responses to a given change in climate greater than later responses to the same magnitude of climate change, as

earlier ablation zones were potentially wider than later ablation zones. To account for this possible volcanic geometrical control on the absolute rate of glacier area change over the past ~120 years, we compare the change in glacier area relative to its prior area.

Glacier length changes are commonly used as a metric to assess glacier relationships to a changing climate (e.g., Oerlemans, 2005; Lillquist and Walker, 2006; Leclercq and Oerlemans, 2012; Leclercq et al., 2014; Roe et al., 2016, 2021; Huston et al.,

2021). Lillquist and Walker (2006) measured glacier length change for White River, Newton-Clark, Eliot, Coe, and Ladd glaciers for the years 1901, 1928, 1938, 1946, 1959, 1967, 1972, 1979, 1984, 1989, 1995, and 2000. Note that Newton-Clark and White River are missing the year 1928 while Ladd is missing the year 1959; these omissions impact the number of observations used to determine the significance of regressions. Lillquist and Walker (2006) also had observations of length change for 2001 for Eliot and Coe; we exclude this year as it is one year removed from 2000 and only exists for two out of

the five glacier records. To this record and along the same flow line we add the change in length up to 2023. In the case of Newton-Clark Glacier, the length is measured in the northern, due-east-facing Newton drainage (Fig. 2). Lillquist and Walker (2006) did not provide uncertainties in their glacier lengths and reported change to 1 m accuracy. Here, we report our length changes at 10 m accuracy.

We compare the cumulative glacier-length change to records of regional climate change. As metrics of regional climate

change, we use U.S. National Oceanic and Atmospheric Administration (NOAA) temperature and precipitation records for Hood River County (Fig. 1) from 1895 to 2023 (U.S. NOAA, 2023). For temperature, we focus on changes in annual and May-October temperature, with the latter being the maximum length of the ablation season. For precipitation, we analyze annual and November-April precipitation, with the latter being the maximum length of the accumulation season. We employ a moving prior-30-year average because this is the U.S. NOAA definition of a climate period. We chose Hood River County

as this is the smallest geographic region that includes these five glaciers and would still have direct weather station measurements within the geographic domain over the period of comparison. Snow telemetry stations on and around Mt. Hood were only installed in the late 1970s, with temperature sensors beginning to work consistently in the late 1980s.

## 3 Results

Mt. Hood has 12 named glaciers recorded on U.S. federal maps; we show the most recent U.S. Forest Service map (2016) in

Fig. 2. We first describe the changes in the 11 remaining glaciers over the last 20 years, Palmer Glacier having ceased to exist prior to this time period. All volcanic age and rock descriptions come from the U.S. Geological Survey Eruption History of Mount Hood, Oregon website (U.S. Geological Survey, 2023). The glacier areas noted for 2015 or 2016 are from





Fountain et al. (2023). We then examine potential factors affecting the magnitude of glacier change. We also investigate the change in area with respect to historic changes since 1907 for seven of these glaciers. Lastly, we test the hypothesis that

changes in five of these glaciers since 1901 are related to regional climate change.

## 3.1 Observed Glacier Change in the 21st Century

Figures 3, 5 and 6 show repeat photographs of the glaciers on Mt. Hood, whereas Fig. 4 has photographs of Coalman Glacier from 2003 and 2021. Figure 7 is a summary of the changes in glacier length and terminus elevation from 2003 to 2023. Figure 8 illustrates glacier area for seven of the glaciers and their sum from 1901/1907 to 2023. Our field measurements are

provided in Table 1.

| Glacier (Multiple Termini) | 2003 Terminus Elevation (m asl) | 2023 Terminus Elevation (m asl) | 20-Year Retreat (m) | 2023 Headwall (m asl) | Accum. Zone Slope (°) | 2023 Area km$^2$ (uncert) |
|---|---|---|---|---|---|---|
| Zigzag | 2350 | 2600 | 710 | 3190 | 55 | 0.101 (0.007) |
| Palmer | gone | | | | | |
| Coalman (E) | 3130 | 3130 | 30 | 3400 | 45 | 0.057 (0.004) |
| Coalman (W) | 3190 | 3230 | 110 | | | |
| White River | 2130 | 2240 | 330 | 3110 | 45 | 0.271 (0.019) |
| Newton-Clark (N) | 2420 | 2480 | 440 | 3410 | 45 | 0.975 (0.068) |
| Newton-Clark (S) | 2320 | 2620 | 680 | | | |
| Eliot | 2050 | 2120 | 310 | 3270 | 70 | 0.912 (0.064) |
| Langille (remain) | | 2520 | | 3290 | 55 | 0.138 (0.010) |
| Langille (upper) | 2070 | gone | | | | |
| Langille (lower) | 1960 | gone | | | | |
| Coe | 1920 | 2090 | 610 | 3280 | 70 | 0.696 (0.049) |
| Ladd | 2100 | 2330 | 770 | 2850 | 60 | 0.235 (0.016) |
| Glisan (upper) | 2140 | gone | | | | |
| Glisan (lower) | 1930 | gone | | | | |
| Sandy | 1890 | 2130 | 520 | 2720 | 50 | 0.631 (0.044) |
| Reid | 2240 | 2350 | 210 | 3010 | 50 | 0.417 (0.029) |

**Table 1: Mt. Hood glacier observations from 2003 and 2023.**





**Figure 3: Repeat photographs for 2003 (left) and 2023 (right) of Mt. Hood's south to east side (Fig. 2). A. Zigzag Glacier across the valley (shot location 45.29275º, -121.80044º). B. White River Glacier (shot location 45.34912º, -121.70219º). C. Southern terminus of Newton-Clark Glacier in the Clark drainage (shot location 45.34747º, -121.66964º). (D) Northern terminus of Newton-Clark Glacier in the Newton drainage (shot location 45.37176º, -121.67088º).**



**Zigzag Glacier** (Fig. 3A) has a southwest aspect with its steep accumulation area (up to 55º) buttressed between two rock
walls that are volcanic bedrock erupted >100 ka (kilo annum). Conversely, the pyroclastic flows from the 1781 eruption of
Mt. Hood created a wide featureless surface onto which the glacier flows in its ablation area. This broad south-facing surface
means Zigzag's terminus is directly exposed to solar radiation. In addition, observations of prevailing winter wind patterns
are observed to redistribute snow from the lower reaches of Zigzag upwards to its accumulation zone. In 2003 there was
minimal debris cover on Zigzag and crevasses were visible in the accumulation zone. The terminus spread broadly across the
pyroclastic flows to a subtle moraine and extended down to ~2350 m asl. By 2023, Zigzag's active terminus had retreated
~710 m and risen in elevation by ~250 m to ~2600 m asl (Fig. 7). The glacier had split into three bodies of ice, all heavily or
entirely debris covered. The uppermost, northernmost, and largest body of ice exhibited still-active crevassing. The two
lower bodies exhibited no actively maintained crevasses, implying stagnation. Other parts of Zigzag noted in 2003 had
transitioned to stagnant ice or disappeared. In 2023, Zigzag's area was $0.101\pm0.007$ km$^2$. In 2015, it was $0.168\pm0.005$ km$^2$,
so the glacier had lost about one-third of its area in the last eight years.

**Palmer Glacier** was first identified as a glacier in 1924, flowing on the same pyroclastic apron as Zigzag Glacier (Nelson,
1924). It was still an active glacier in 1981 as its maximum ice thickness reached ~60 m (Driedger and Kennard, 1986). As
of 2003, it was no longer an active glacier, only a perennial snowfield over stagnant ice without crevasses or evidence of
effluent glacial flour. By 2023, this snowfield was now seasonal (Fig. 2, 3), disappearing in late September to October
despite proactive efforts by Timberline Lodge Ski Area to farm and store snow for summer ski operations. Once this
snowfield melts in late summer, small stagnant ice from the once-active glacier remained under debris and ski-area-related
trash.

**Coalman Glacier** (Fig. 4) is south facing, occupies a collapsed lava dome (erupted ~1.5 ka) and is buttressed by cliffs to the
east (erupted 12-30 ka) and west (erupted >100 ka). Coalman's flow is split by a rock promontory into two lobes: one
descending to the southwest towards Zigzag Glacier and the other descending to the southeast towards White River Glacier
(Fig. 4A). Large fumaroles formed in the late 1800s around the rock promontory and both are active to this day, likely
influencing the glacier (Lillquist and Walker, 2006). Indeed, Coalman was part of the accumulation zones of White River
and Zigzag glaciers in the 1800s, separating from White River Glacier by 1912 and later from Zigzag Glacier (Lillquist and
Walker, 2006). The accumulation area slopes up to 45º and extended to the summit of Mt. Hood until the late 1990s. In
2003, the western lobe flowed down to ~3190 m asl while the eastern lobe reached ~3130 m asl, terminating in a lake near a
fumarole (Fig. 4A). By September 2021, our last visit to the summit when photography was possible, what remained was an
isolated ice mass resting on the steep slope of the ridge to the west; the highest remaining accumulation area on a well-
buttressed slope reduced to dark firn with debris along the edges and old crevasses filling with debris; a single tiny patch of
dead ice on the large gently-sloped summit area (Fig. 4B, C); and a thinning area of active ice just above the aforementioned
lake to the east (Fig. 4A). In 2023, repeat photography of the glacier was not possible due to Coalman Glacier being in a



cloud. The glacier's eastern terminus had receded ~30 m since 2003 with no significant change in elevation while the western terminus had retreated ~110 m and risen ~40 m in elevation to ~3230 m asl. We mapped the glacier area as 0.057±0.004 km² in 2023, 40% smaller than its 2016 area of 0.099±0.001 km² just seven years before. Much of this area loss occurred along the western lobe that is exposed to more direct solar radiation than the eastern lobe.

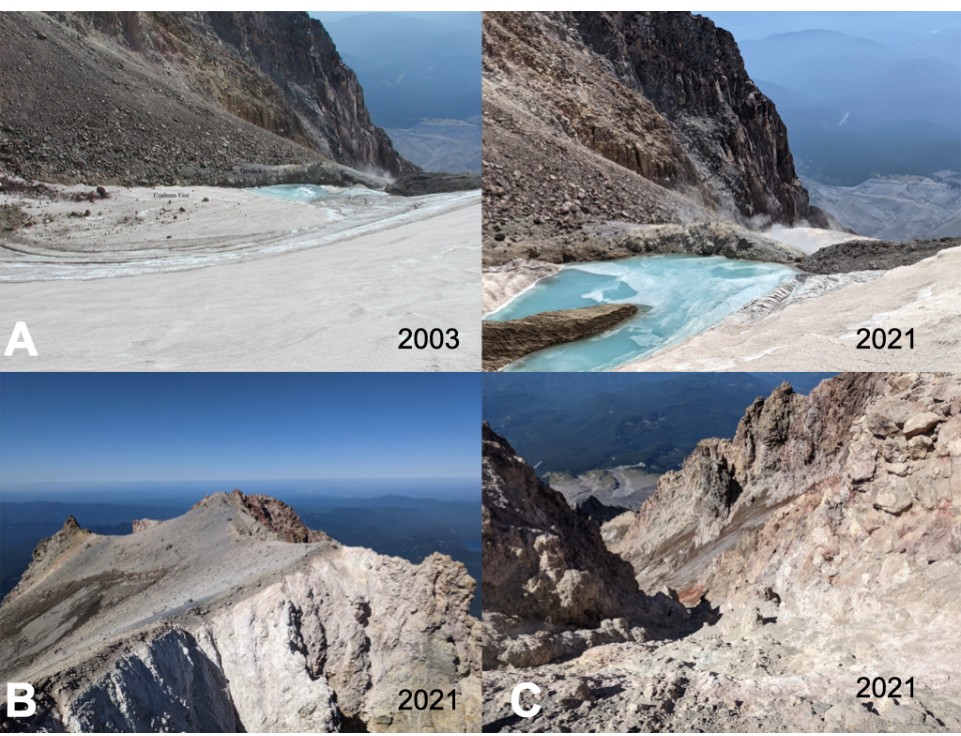


**Figure 4: Photographs of Coalman Glacier from 2003 and 2021. A. The eastern lobe in 2003 and in 2021. B. Remnant ice in 2021 that used to be the upper part of Coalman's accumulation zone in 2021. C. Stagnant ice in 2021 that was the upper part of Coalman's accumulation zone.**

**White River Glacier** (Fig. 3B) faces due south so the whole glacier receives direct solar radiation. Before 1912, the glacier extended up into Coalman Glacier (Lillquist and Walker, 2006). Since then, the glacier's accumulation area has begun below a fumarole and flows between the edge of the pyroclastic flows to the west and a rock cliff to the east with a maximum slope of ~45º. At lower elevations, these pyroclastic flows created a plain upon which the White River terminus spread in the early 1900s (Fig. 1). In 2003, the glacier descended to an elevation of ~2130 m asl, with debris cover on the lowest 100 m of the

terminus. The terminus retreated ~330 m and rose in elevation by ~110 m to ~2240 m asl in the last 20 years. As of 2023, the glacier had experienced significant thinning and lost most large crevasses near its terminus, yet it had maintained similar debris coverage. Farther up, White River had large crevasses especially in the upper accumulation area. In 2023, it had an area of 0.271+0.019 km². In 2015, the glacier's area was 0.287±0.046 km².



**Newton-Clark Glacier** (Fig. 3C, D) faces east with a broad and up to 45º accumulation zone that has minimal buttressing to
the north and south. The glacier flows into two drainages: Clark to the southeast (Fig. 3C) and Newton to the east (Fig. 3D),
with the former having greater exposure to direct solar radiation than the latter. The 2003 margin in the Clark drainage
reached down to ~2320 m asl while the margin in the Newton drainage extended down to ~2420 m asl. Debris cover on the
terminus was moderate, but with evidence of a recent landslide onto the glacier from the headwall. In 2023, extensive debris
coated the Newton-Clark Glacier terminus in the Newton drainage. We observed a landslide onto the glacier while field
mapping, with additional rockfall occurring every 5 to 10 minutes. The new landslide covered a large portion of the northern
uppermost area of the glacier and added to the significant debris coverage that dominates most of the glacier's surface. In the
last 20 years, the active ice margin in the Clark drainage retreated ~680 m and rose to ~2620 m asl, with only stagnant ice
remaining. The southeast active ice margin was now on the ice divide between the two drainages, which itself retreated ~110
m. The ice margin in the Newton drainage retreated ~440 m, with the terminus rising to ~2580 m asl. Newton-Clark's active
ice area was 0.975±0.068 km$^2$ in 2023 while its area was mapped as 1.134±0.181 km$^2$ in 2015.

**Eliot Glacier** (Fig. 5A) faces northeast and flows in a glacial trough that is reinforced by an extensive moraine that dates to
at least the 1700s (Lawrence, 1948). The accumulation zone approaches 70º on a 12-13 ka dacite surface that shades the
zone for much of the day. The terminus is heavily debris-covered due to rock fall and eolian and englacial debris (Lundstrom
et al., 1993; Lillquist and Walker, 2006; Jackson and Fountain, 2007). The active glacier margin under debris terminated at
~2050 m asl with stagnant debris-covered ice found down to ~1900 m asl in 2003. By 2023, the active and stagnant ice
margins had retreated ~310 m and ~300 m, respectively, with the active glacier margin rising ~70 m to ~2120 m asl. The
glacier had lost significant large crevasses and seracs that in 2003 were flowing over rock promontories and cliffs, which
were now exposed. The terminus had thinned significantly in those 20 years. The glacier's 2023 area was 0.912±0.064 km$^2$
versus 1.086±0.174 km$^2$ in 2016.

**Langille Glacier** (Fig. 5B) faces north-northeast, lies between Eliot and Coe glaciers, and has a steep accumulation zone
before flowing over a broad slope. By 2003, the lower portions of the glacier had separated from the accumulation zone and
broken into four separate ice masses. Two of them still had crevasses, terminating at ~2070 m asl and ~1960 m asl. By 2023,
almost all of the ice in these four ice bodies had melted away. The remaining small patches of stagnant ice were largely
debris-covered. The active glacier now covered 0.138±0.010 km$^2$. In 2015, Fountain et al. (2023) noted these stagnant ice
portions as perennial snowfields, a classification that is mostly no longer applicable, and they placed Langille's active area at
0.233±0.037 km$^2$.



**Figure 5: Repeat photographs from 2003 (left) and 2023 (right) of Mt. Hood's northeast to north side (Fig. 2). A. Eliot Glacier (shot location 45.38891º, -121.66957º). B. Langille and Coe Glaciers (shot location 45.39459º, -121.69982º). C. Coe Glacier (shot location 45.39701º, -121.69092º). (D) Ladd Glacier (shot location 45.39718º, -121.70424º).**



**Coe Glacier** (Fig. 5B, C) faces due north and flows in a glacier trough that is buttressed by its moraines like Eliot Glacier. Also similar to Eliot, Coe has a very steep accumulation zone (up to 70º). However, Coe's terminus has less debris cover than Eliot's terminus. In 2003, Coe's lowest ice fall was still active with ice flowing down to ~1920 m asl. Its debris-covered

stagnant ice extended to ~1780 m asl. By 2023, the ice fall had melted away with the active ice terminating above the now-exposed cliff at ~2090 m asl. All the ice below this cliff was now stagnant and debris covered. While crevassing remains significant in 2023, it had reduced since 2003, with smaller and fewer crevasses. Ice thinning had also exposed larger rock cliffs. The active ice margin retreated ~610 m while the stagnant ice melted back ~570 m. Coe's area was $0.696\pm0.049$ km$^2$ in 2023 whereas the glacier covered $0.779\pm0.125$ km$^2$ in 2016.

**Ladd Glacier** (Fig. 5D) has a north-northwest aspect, and a steep accumulation zone (up to 60º) adjacent to Coe's but not reaching as high. Ladd flows in another glacial trough parallel to Coe until making a bend westward at an andesite spur that erupted ~50 ka. Thus, Ladd's lower reaches do not have the shading from solar radiation afforded Eliot and Coe by the summit of Mt. Hood. Below the spur, Ladd is buttressed to the north by a prehistoric moraine but spreads broadly to the south. In 2003, the active glacier terminus had high amounts of debris cover with three sub-lobes spread out to elevations of

~2120 m asl (north), ~2110 m asl (south), and ~2100 m asl (central), with the central lobe having seracs. In the last 20 years, Ladd had experienced the most ice loss of any glacier on Mt. Hood, receding ~770 m to a single lobe at ~2330 m asl above a previous icefall. More notably, the glacier had disconnected from its uppermost accumulation zone due to ice thinning over another ice fall. The active ice remaining in Ladd covered $0.235\pm0.016$ km$^2$ in 2023 while it spanned $0.347\pm0.055$ km$^2$ in 2016; the glacier lost about a third of its area in seven years.

**Glisan Glacier** (Fig. 6A, B) had a northwest aspect, a shallow and relatively short accumulation zone, and minimal buttressing to the south. By 2003, the glacier had broken up into four ice masses, only two of which had crevasses: one terminating at ~2140 m asl (Fig. 6B) and the other at ~1930 m asl (Fig. 6A). Both of these ice masses had previously formed distinct lateral and terminal moraines from which they were retreating in 2003. In 2023, the remnant with the higher terminus was nearly gone while the other with the lower terminus had stagnated. One of the 2003 stagnant ice masses was

completely gone while the other remained. For 2015, Fountain et al. (2023) documented the glacial remnant against the bedrock ridge as still active with an area of $0.041\pm0.006$ km$^2$ and classified the other 2003 remnant as a perennial snowfield. Both these classifications were out-of-date as of 2023 and Glisan Glacier no longer existed.



Figure 6: Repeat photographs from 2003 (left) and 2023 (right) of Mt. Hood's west side (Fig. 2). A. Glisan Glacier remnant looking south (shot location 45.39531º, -121.71777º). B. Glisan Glacier remnants looking east (shot location 45.39093º, -121.72249º). C. Sandy Glacier (shot location 45.39477º, -121.73124º). (D) Ladd Glacier (shot location 45.37559º, -121.74100º).



**Sandy Glacier** (Fig. 6B) faces west-northwest in a broad valley between two andesitic ridges (erupted 50-100 ka). The slope

of its accumulation area exceeds 50º on a headwall before shallowing to <10º. This glacier is the only one on Mt. Hood that

has an ice cave system. In 2003, the glacier had three terminal lobes, the lowest of which reached ~1890 m asl and had two

ice caves. One of the caves opened higher up on the glacier and may have communicated with an ice cave system discovered

in the accumulation zone after our observation. The main and lowest terminus in 2003 was covered by what appeared to be a

rock avalanche but the rest of the glacier was largely free of debris. In 2023, Sandy's main terminus hae retreated ~520 m to

~2130 m asl. Two ice caves still existed. Debris cover had greatly increased on the glacier, blanketing even large parts of its

accumulation zone. The glacier had an area of $0.631\pm0.044$ km$^2$ in 2023. In 2016, the glacier spanned $0.751\pm0.120$ km$^2$.

**Reid Glacier** (Fig. 6C) faces west in another broad valley between two bedrock ridges erupted >100 ka. The accumulation

zone has a slope upwards of 50º while its terminus has in the past flowed over a large cliff that could limit further downslope

expansion of the glacier. The southern ridge provides shading to the glacier's terminus from direct solar radiation for much

of the day. In 2003, most of the terminus resided above the cliff while three areas of the terminus spilled over the cliff at

~2240 m asl. The central portion of the glacier still connected to recently stagnated (as evidenced by down-wasting

crevasses) debris-covered ice below the cliff reaching to ~1920 m asl. In 2023, the glacier terminus resided entirely above

the cliff, and the section of the terminus to the north that spilled over the cliff as an icefall in 2003 was stagnant. Some

heavily debris-covered stagnant ice resided below the cliff band. Reid had experienced significant thinning from 2003 to

2023, with more debris cover from rockfall and landslides, including parts of its accumulation zone. It had  retreated ~210 m

since 2003 with the terminus rising to ~2350 m asl. The glacier had a 2023 area of $0.417\pm0.029$ km$^2$ versus $0.469\pm0.005$ km$^2$

in 2015.

**Seven glaciers** (White River, Newton-Clark, Eliot, Coe, Ladd, Sandy, and Reid) have area records that extend back to 1907

(Jackson and Fountain, 2007). From 2000 to 2023, these glaciers lost a combined ~2.8 km$^2$ ($7.11\pm0.21$ km$^2$ to $4.3\pm0.12$ km$^2$)

or ~40% of their 2000 area (Fig. 8). When related to their 1907 combined areas, they have lost ~5.7 km$^2$ ($9.98\pm0.37$ km$^2$ to

$4.3\pm0.12$ km$^2$) or ~60% of their 1907 area. For context, from 1907 to 2000, these glaciers lost ~2.9 km$^2$ ($9.98\pm0.37$ km$^2$ to

$7.11\pm0.21$ km$^2$) or ~30% of their 1907 area. Between 2000 to 2023, the total area lost by these seven glaciers about doubled

what was lost between 1907 to 2000.






**Figure 7: Change in Mt. Hood glacier termini locations from 2003 to 2023. A. Change in length relative to aspect on Mt. Hood; ZZ = Zigzag, C = Coalman, WR = White River, NC = Newton-Clark with its two termini labeled, E = Eliot, and R = Reid. B. Change in elevation relative to aspect on Mt. Hood. C. Change in length versus elevation with glacier termini labeled. Dashed line is significant regression ($r^2$ = 0.86, p = 0.00).**






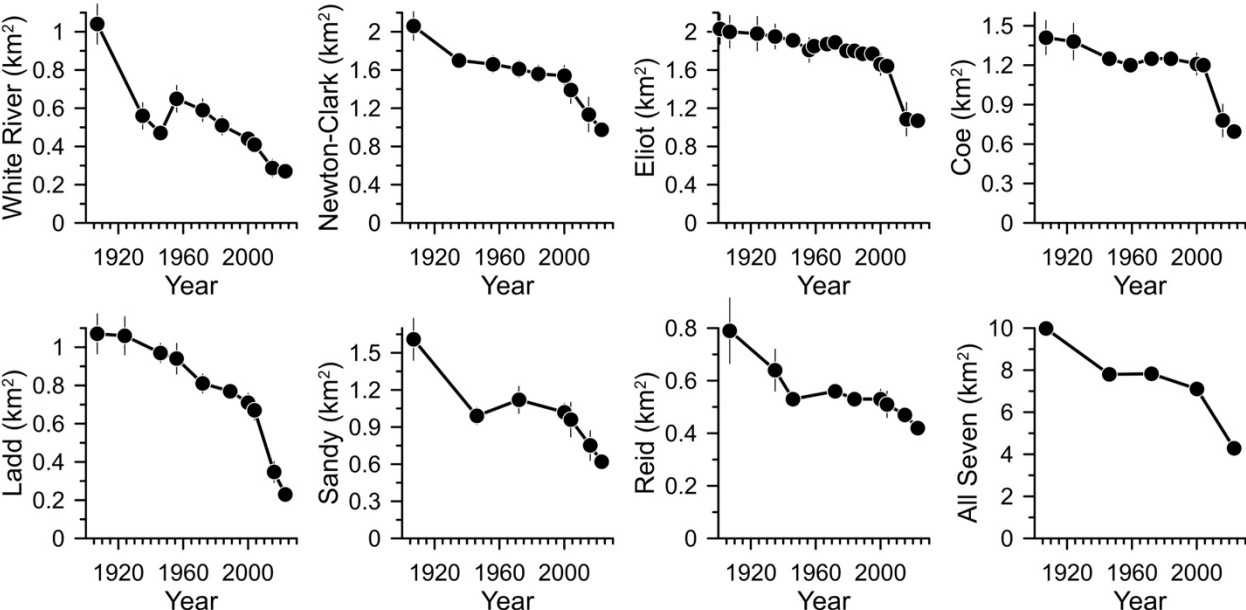

**Figure 8: Glacier area from 1907 (or 1901 for Eliot Glacier) to 2004 from Jackson and Fountain (2007), 2015 or 2016 from Fountain et al. (2023), and 2023 (new data), with combined area noted in the lower right. Uncertainty noted by vertical bars (note that the uncertainty on the combined record is smaller than the symbol size).**

### 3.2 Potential factors in glacier retreat

All glaciers on Mt. Hood retreated in the last 20 years with a rise in the elevation of their termini. We find a strong relationship between the magnitude of terminus retreat and the degree of terminus-elevation rise (Fig. 7C), with an $r^2$ of 0.86 (p < 0.00). We now examine potential controls that could explain the different magnitude of terminus retreat between the glaciers. We consider glacier aspect, accumulation-area slope, highest point of the accumulation zone, valley shape and potential buttressing, debris cover, and substrate, as well as glacier-specific influences (e.g., fumaroles) and the amount of 20th century retreat.

The degree of terminus retreat and rise in elevation does not corelate with aspect (Fig. 7A, B). There is also no significant relationship between the slope of glacier accumulation area (p = 0.28) or the highest elevation of the accumulation area (p = 0.33) and the magnitude of terminus retreat. While difficult to quantify buttressing, we do not see a correlative pattern of glaciers flowing over a broad surface with minimal buttressing (Zigzag, Newton-Clark) receding more than glaciers in wide valleys (Sandy, Ladd), narrow valleys (White River, Reid), or very narrow valleys (Coe, Eliot). Debris cover may influence the magnitude of glacier retreat but does not correlate with the magnitude of retreat. The glaciers with the highest debris cover (Eliot, Coe) have retreated around the average of Mt. Hood's glaciers while a third glacier with heavy debris cover (Ladd) has, for other reasons, retreated the most (Figure 8C). Furthermore, substrate type and age (i.e., centuries-old pyroclastic flows versus tens of millennia old andesite or dacite rock) does not appear to be a factor, because glaciers on





older rock (Coe, Eliot, Ladd) versus younger volcanic sediment (Zigzag, White River) have similar magnitudes of margin
recession.

When comparing adjacent glaciers, topographic and other factors could explain different amounts of retreat. Sandy Glacier
retreated ~310 m more than Reid Glacier, which could be due to Sandy having an ice cave system that likely enhances
ablation; Reid having greater shading and buttressing; and Reid terminating at a cliff that impeded down-valley extension
thereby reducing its initial retreat to a climate perturbation. The existence of fumaroles since at least 1907 certainly have
influenced the behavior of Coalman Glacier's two lobes (Lillquist and Walker, 2006), which had the least amount of retreat
on Mt. Hood (30 and 110 m). In the absence of this volcanic heat, Coalman Glacier might still be connected to White River
and Zigzag glaciers as part of their accumulation zones, and thus not have retreated at all. Coe and Eliot are similar glaciers,
but Eliot has more debris cover and shading than Coe, which probably explains Coe's greater retreat. Ladd Glacier's loss of
part of its accumulation zone and the westward bend of its ablation zone must have contributed to the largest retreat of any
glacier on Mt. Hood in this century.

### 3.3 Decadal Rate of Area Loss for Seven Glaciers

Figure 9 shows the absolute rate of change in aerial extent for these seven glaciers while Fig. 10 illustrates the relative rate of
change in aerial extent. From 2000 to 2023, White River Glacier lost area at a rate of 0.066-0.081 $km^2$ decade$^{-1}$, which is a
little less than half the maximum rate of area loss that occurred from 1907 to 1946 of 0.127-0.165 $km^2$ decade$^{-1}$. However,
when assessed as the rate of loss relative to prior extent, the most recent 23 years of retreat at 15.0-18.4 % decade$^{-1}$ was
probably faster than the 1907-1946 rate of retreat at 12.2-15.9 % decade$^{-1}$. Reid Glacier also retreated at the second fastest
rate of 0.043-0.053 $km^2$ decade$^{-1}$ for 2000-2023, which is slightly slower than its 1907-1946 rate of 0.056-0.078 $km^2$ decade$^{-1}$. Reid's 2000-2023 relative retreat rate was about the same at 8.1-9.9 % decade$^{-1}$ as its 1907-1946 rate of 7.1-9.8 % decade$^{-1}$.
$^{-1}$. Sandy Glacier's 2000-2023 rate of area loss of 0.157-0.191 $km^2$ decade$^{-1}$ overlaps with its 1907-1946 rate of 0.138-0.179
$km^2$, but its relative retreat rate of 15.4-18.7 % decade$^{-1}$ for 2000-2023 is faster than the 8.6-11.1 % decade$^{-1}$ for 1907-1946.

The rates of retreat from 2000 to 2023 for Newton-Clark (0.221-0.270 $km^2$ decade$^{-1}$; 14.4-17.5 % decade$^{-1}$), Eliot (0.231-
0.282 $km^2$ decade$^{-1}$; 13.9-17.0 % decade$^{-1}$), Coe (0.201-0.246 $km^2$ decade$^{-1}$; 20.3-26.6 % decade$^{-1}$), and Ladd (0.188-0.229
$km^2$ decade$^{-1}$; 26.5-32.3 % decade$^{-1}$) glaciers are unmatched since 1907 at the common resolution of these records. For
comparison, the prior rates of maximum retreat for these glaciers were for Newton-Clark from 1907-1946 at 0.089-0.107
$km^2$ decade$^{-1}$ (4.3-5.2 % decade$^{-1}$), for Eliot from 1972-2000 at 0.076-0.088 $km^2$ decade$^{-1}$ (4.0-4.7 % decade$^{-1}$), for Coe from
1907-1946 at 0.037-0.045 $km^2$ decade$^{-1}$ (2.6-3.2 % decade$^{-1}$), and for Ladd from 1946-1972 at 0.057-0.66 $km^2$ decade$^{-1}$ (5.8-
6.9 % decade$^{-1}$).

When the glaciated area of these seven glaciers is combined, the aerial retreat over the last 23 years is the fastest since 1907.
From 2000 to 2023, these seven glaciers lost area at 1.179-1.280 $km^2$ decade$^{-1}$, which is 2.0-2.4 times the rate from 1907 to





1946 of 0.536-0.582 km$^2$ decade$^{-1}$. The relative rate of change in glacier area is 16.6-18.0 % decade$^{-1}$ for the last 23 years, which is 2.9-3.3 times faster than the relative rate for 1907-1946 of 5.4-5.8 % decade$^{-1}$.

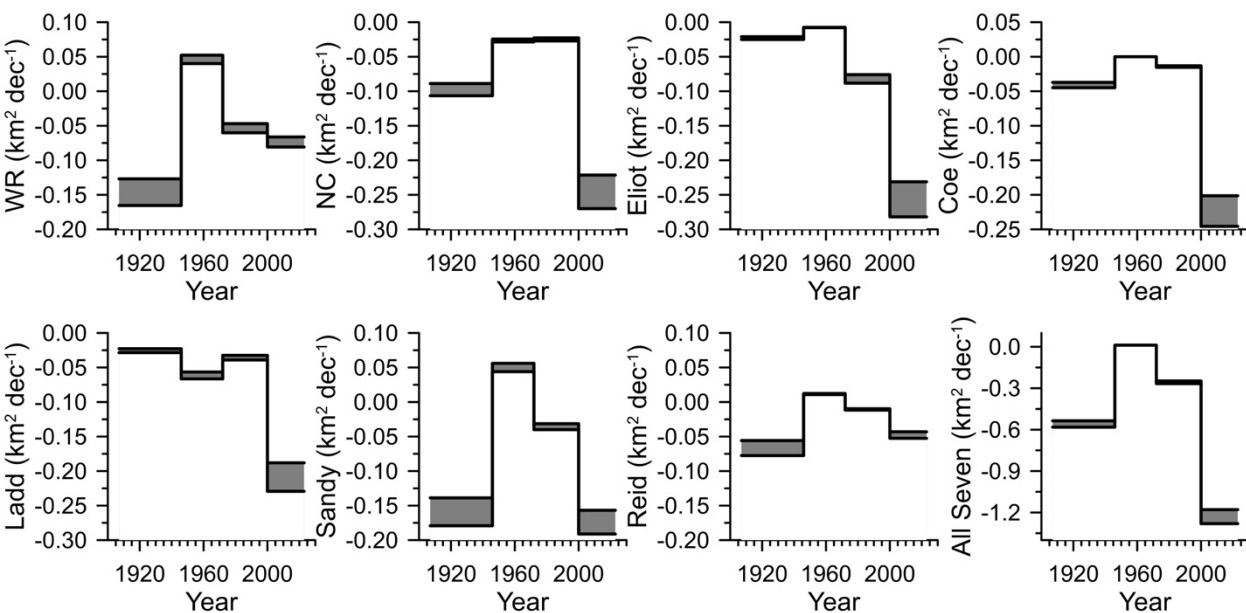

**Figure 9: Rate of area change (km$^2$ dec$^{-1}$ where dec = decade) calculated from Jackson and Fountain (2007) (1907-2000) and 2023 observations, with combined rate in the lower right. Gray shading shows the propagated uncertainty. WR = White River and NC = Newton-Clark.**

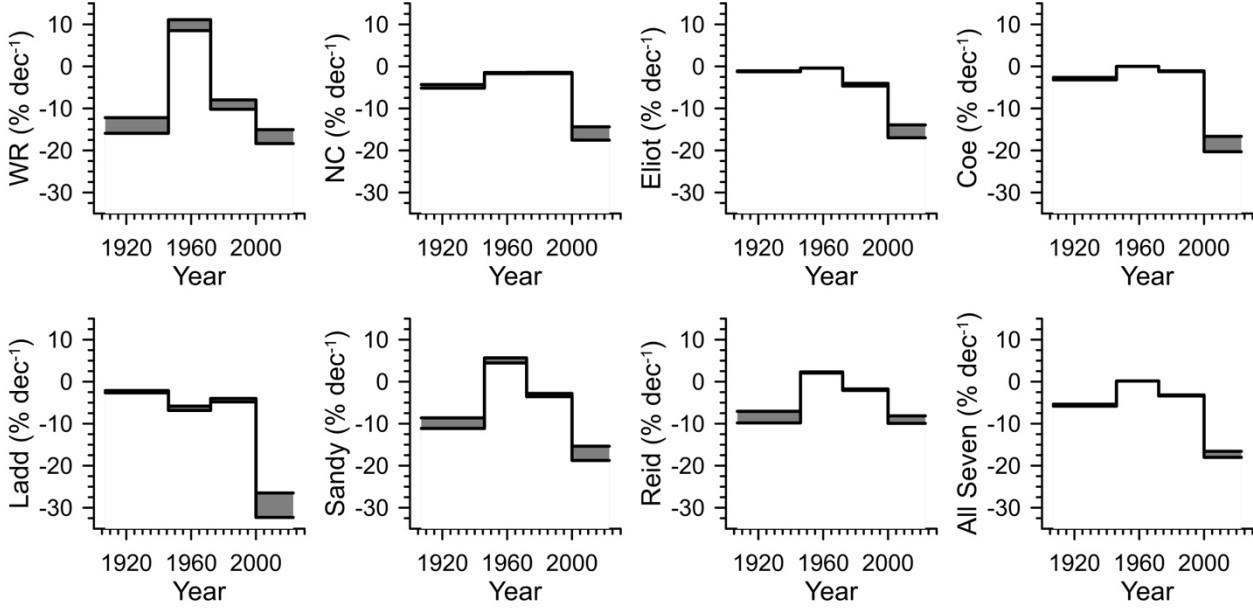

**Figure 10: Relative of area change with respect to preceding ice area expressed (% dec$^{-1}$ where dec = decade) calculated from Jackson and Fountain (2007) (1907-2000) and 2023 observations, with combined rate in the lower right. WR = White River and NC = Newton-Clark.**





### 3.4 Glacier-Climate Relationships

Figure 11 shows the change in glacier length relative to the changes in prior-30-year average temperature (annual and May-October) and total precipitation (annual and November-April). Figure 12 contains the regressions of glacier length against the prior-30-year average temperature and precipitation; Table 2 documents the significance of the results. In all cases, we

find significant relationships between glacier length and annual and May-October temperature change, with $r^2$ ranging from 0.42 (Newton-Clark annual, p = 0.03) to 0.91 (Eliot annual, p < 0.01). Conversely, there is no significant relationship between glacier-length changes and total or November-April precipitation, with $r^2$ ranging from 0.02 (White River annual, p = 0.71) to 0.25 (Coe annual, p = 0.10).

| Glacier | Annual T (1901-2023) $r^2$, p | MJJASO T (1901-2023) $r^2$, p | Annual T (1901-2000) $r^2$, p | MJJASO T (1901-2000) $r^2$, p | Annual P (1901-2023) $r^2$, p | NDJFMA P (1901-2023) $r^2$, p | Annual P (1901-2000) $r^2$, p | NDJFMA P (1901-2000) $r^2$, p |
|---|---|---|---|---|---|---|---|---|
| White River | 0.63, <0.01 | 0.67, <0.01 | *0.10, 0.37* | *0.10, 0.38* | *0.02, 0.71* | *0.02, 0.70* | *0.16, 0.25* | *0.09, 0.41* |
| Newton-Clark | 0.42, 0.03 | 0.58, 0.01 | *0.01, 0.78* | *0.03, 0.66* | *0.04, 0.56* | *0.08, 0.41* | *0.24, 0.15* | *0.25, 0.14* |
| Eliot | 0.91, <0.01 | 0.86, <0.01 | 0.77, <0.01 | 0.67, <0.01 | *0.09, 0.34* | *0.09, 0.34* | *0.09, 0.38* | *0.15, 0.24* |
| Coe | 0.87, <0.01 | 0.87, <0.01 | 0.55, 0.01 | 0.48, 0.02 | *0.25, 0.10* | *0.17, 0.18* | *0.56, 0.01 | *0.49, 0.02 |
| Ladd | 0.82, <0.01 | 0.73, <0.01 | 0.70, <0.01 | 0.56, 0.01 | *0.06, 0.47* | *0.05, 0.51* | *0.03, 0.64* | *0.05, 0.54* |

**Table 2: Regression $r^2$ and p from Fig. 12. Italics indicates non-significant results. An asterisk identifies a significant correlation that is opposite of an expected glacier-climate relationship.**

We test the influence of our 2023 observations on the statistically significant relationships between glacier length and temperature by conducting the same analysis but for 1901 to 2000. For Eliot, Coe, and Ladd glaciers, the strength of the relationship decreases for annual and May-October temperature but remains statistically significant (p = < 0.01 to 0.02). In

contrast, White River and Newton-Clark glaciers do not have statistically significant relationships with either annual or May-October temperature. A similar test with annual and November-April precipitation yields significant relationships only for Coe Glacier (p = 0.01, 0.02), where greater glacier retreat is associated with higher annual (p = 0.01) and November-April (p = 0.02) precipitation (Fig. 12), which is opposite of the usual relationship between glacier retreat and lower precipitation.







Figure 11: Hood River County climate change (NOAA) and Mt. Hood glacier length change (Lillquist and Walker, 2006). A. Prior 30-year average for total annual (dark blue) and November-April (light blue) precipitation relative to 1895-1924 average. B. Prior 30-year average for mean annual (dark red) and May-October (pink) temperature relative to 1895-1924 average. C. Change in glacier length from 1901 to 2023 for White River (purple), Newton-Clark (green), Eliot (orange), Coe (black), and Ladd (red) Glaciers.








**Figure 12: Regression of Mt. Hood glacier-length change (Lillquist and Walker, 2006) against (from left to right) NOAA Hood River County prior 30-year average mean annual temperature, mean May-October temperature, total annual precipitation, and total November-April precipitation. Dashed lines are significant (p < 0.05) regressions for 1901-2023 while gray lines are significant (p < 0.05) regressions for 1901-2000.**





## 4 Discussion

Over the last 20 years, every glacier on Mt. Hood has reduced in length and lost area, with a rise in terminus elevation. Such clear glacier retraction stands at odds with the Menounos et al. (2019) finding of near-neutral mass change for glaciers on Mt. Hood from 2000 to 2018. We note that their study lacked ground truthing of their geodetic measurements in Oregon, which combined with the high and growing debris cover on many of Mt. Hood's glaciers, could explain this disagreement. Furthermore, Florentine et al. (2024) found that not accounting for glacier area change when measuring geodetic mass balance reduces the mass-balance-change signal in western North American glaciers, with greater area loss leading greater mass-balance bias. Here, our field observations clearly show large-scale glacier loss, consistent with the 21st-century retreat of glaciers around the globe (Zemp et al., 2015) and in agreement with the cumulative mass loss of benchmark glaciers to the north of Mt. Hood in Washington State (Pelto, 2023).

In the last 20 years, Glisan Glacier on Mt. Hood ceased to be a glacier, joining Palmer Glacier in this former-glacier status. We find that two other glaciers are nearing this status: Zigzag and Coalman. Zigzag Glacier is only distinguishable from stagnant ice due to the presence of a few crevasses that have yet to start melting in on themselves (i.e., movement is keeping them open). Coalman Glacier at ~0.06 km$^2$ is now below the area threshold used by the U.S. Geological Survey to distinguish active glaciers of 0.1 km$^2$ (Fagre et al., 2017) but still has one active crevasse. At ~0.1 km$^2$, Zigzag Glacier is also near this U.S. Geological Survey glacier demarcation. With these two glaciers losing 30-40% of their area in the last seven to eight years, we suggest that in the near future they will stagnate and join Palmer and Glisan as former glaciers. Ladd Glacier is also retreating rapidly, losing about a third of its area in the last seven years. If this rapid retreat continues, we predict that Ladd Glacier will be the next glacier, after Zigzag and Coalman, to reach stagnation on Mt. Hood.

The rate of area loss in the last 23 years is the greatest in the last ~120 years for four of the seven glaciers on Mt. Hood, and another two glaciers have retreat rates near their 1907-1946 rates (Fig. 9) despite the potential influence of volcano geometry that would favor earlier faster rates (Fig. 1). Only White River Glacier had a significantly faster absolute rate of area loss from 1907 to 1946 (Fig. 9). This faster rate was likely facilitated by volcanic activity between 1853 and 1907 that separated White River Glacier from Coalman Glacier, with the reduction in accumulation area driving greater White River retreat (Lillquist and Walker, 2006).

The sum of the retreat rates of the seven Mt. Hood glaciers over the first 23 years of the 21st century stands out as unparalleled since at least the beginning of the 1900s (Fig. 9). In the 20th century, the maximum absolute and relative rate of area loss for these glaciers occurred 1901-1947. This 1907-1946 interval of retreat corresponds with a warming-drying period in the Pacific Northwest that is attributable to human greenhouse gas emissions (Marvel et al., 2019). Furthermore, the stabilization or even slight advance of Mt. Hood's glaciers 1946-1972 (Fig. 8, 9) coincides with a cooler-wetter period in




the Pacific Northwest that was in response to human-aerosol emissions counteracting human-greenhouse gas emissions (Marvel et al., 2019).

Because of the statistically significant relationship between the change in glacier length for five of Mt. Hood's glaciers and the change in regional temperature, we argue that the retreat of glaciers over the last ~120 years reflects regional climate warming (Roe et al., 2016, 2021). We do not find a similar relationship with precipitation. Interestingly, we did find

significant relationships between Eliot, Coe, and Ladd glacier lengths and prior 30-year average temperatures for the 20th century while Lillquist and Walker (2006) did not. We attribute this difference, at least partly, to how the climate data were analyzed. Lillquist and Walker (2006) used a 5-year running average for both temperature and precipitation, which would mean the glacier's current length is only related to the current year and two prior years of weather, with the coming two years of weather having yet to occur. In contrast, we examine the change in the prior 30-year average that reflects the change

in climate as defined by the U.S. NOAA.

However, we did not find a significant relationship for White River and Newton-Clark glaciers with temperature or precipitation over the 20th century. White River's early 20th-century retreat clearly had a non-climatic cause of glacier change from the partial loss of its accumulation zone due to volcanic activity (Lillquist and Walker, 2006). Newton-Clark's main body in the Newton drainage terminated at a cliff for much of the 20th century. This cliff might have limited its down

valley extent, reducing its response to a warming climate until the climate warmed sufficiently to drive retreat from the cliff as is now the case. Nevertheless, both glaciers have a significant temperature relationship once the 21st century changes are included.

Frans et al. (2016) modeled the evolution of Newton-Clark, Eliot, Coe, and Ladd glaciers from 2000 to the end of the century under intermediate (RCP4.5) and high (RCP8.5) emission scenarios, with 10 simulations per emission scenario.

From 2000 to 2025, they simulated a 10-30% loss in glacier area regardless of emission scenario. In contrast, we find that the area of these four glaciers changed from $5.12\pm0.19$ km$^2$ in 2000 to $2.97\pm0.11$ km$^2$ in 2023, losing 30-50% of their area. This means that glacier recession on Mt. Hood in the 21st century has probably proceeded at a faster rate than simulated for even the highest emission scenario. This also raises concerns over the persistence of Mt. Hood glaciers in the 21st century. Frans et al. (2016) modeled continued glacier presence on Mt. Hood in 2100 even under RCP8.5 forcing. However, the observed

glacier retreat in the last 20 years has equaled to exceeded by 67% the maximum loss simulated under RCP8.5 emissions over the same time period.

We hypothesize that the observed faster retreat of glaciers on Mt. Hood relative to their simulated retreat may, in part, be due to climate-related weather phenomena that were not included in the model forcing of Frans et al. (2016). In 2015, Oregon experienced the lowest spring snowpack ever recorded (Mote et al., 2016), which was followed by the warmest summer ever

recorded (U.S. NOAA, 2023). Such extreme events are absent from the Frans et al. (2016) forcing and may have



significantly impacted glacier mass balance (Pelto, 2018). Likewise, the June 2021 Pacific Northwest heatwave was one of the six most extreme events ever recorded on the planet (Thompson et al., 2022), which would have negatively impacted glacier mass balance (Pelto et al., 2022). In 2020, more forest area burned in the Oregon Cascades than all 36 prior years of record combined (Abatzoglou et al., 2021). Since 2015, fire-smoke particulates have been rising in Oregon (Burke et al.,
2023). These light-absorbing particulates from these fires accumulate on Oregon glaciers, where they reduce albedo and increase ablation (Kaspari et al., 2015; Allgaier et al., 2022). Given this under projection of glacier change on Mt. Hood and analogous snow-drought and heatwave impacts (Mote et al., 2016; Philip et al., 2022; Thompson et al., 2022) along with a rise in fire-smoke particulates (Neff et al., 2012; Kaspari et al., 2020), we suggest that such hindsight assessments could be conducted for glaciers in Washington's Olympic, Central Cascade, and North Cascade ranges and all the glaciers of British
Columbia where similar simulations have been conducted (Clarke et al., 2015; Frans et al., 2018).

The rapid glacier retreat in the 21st century that we document here raises questions over the timing of peak meltwater for glaciers on Mt. Hood. Huss and Hock (2018) simulated peak melt water occurring between 2010 and 2020 using the intermediate RCP4.5 emission scenario. Frans et al. (2018) modeled peak melt water for Mt. Hood glaciers as occurring between 2010 and 2050 with a median of 2030 for RCP4.5 and RCP8.5. Note that this timing of peak meltwater is based on
the same simulations of Frans et al. (2016) that, with the now-afforded hindsight, under-projected retreat of Mt. Hood glaciers. While our glacier-area-change data do not directly address the timing of peak meltwater, we suggest that the rapid retreat we document here implies that peak meltwater has already occurred on Mt. Hood, similar to the simulations of Huss and Hock (2018).

Our finding that the Frans et al. (2016, 2018) model under simulates recent glacier recession on Mt. Hood is of vital import
to downstream communities. The U.S. Bureau of Reclamation conducted a Hood River Basin Study that utilized the same model as Frans et al. (2016, 2018). This federal study projected only a 1-4% decline in Mt. Hood glacier area with increased meltwater discharge through 2060, meaning peak water had yet to be reached in these simulations (U.S. Bureau of Reclamation, 2015). We argue that this study, which underlies water management plans for the Hood River Basin (Thieman et al., 2021), contains overly optimistic simulations of future glacier retreat, and water resource managers that rely on this
federal study are ill-prepared for future late-summer declining stream flow and rising instream temperatures.

Thieman et al. (2021) noted that glacier disappearance in the Hood River Basin would be catastrophic for salmon populations. Our findings imply this this could happen sooner than is projected in the current Hood River Basin plans that have 2100 as the earliest possible year for deglaciation. The cold instream temperatures of Hood River are due to glacier meltwater and afford the river its diversity in salmon, steelhead, and bull trout (Thieman et al., 2021). Deglaciation of the
river basin in this century would upset this river diversity as late-summer instream temperatures permanently warm. Rapid glacier recession also imperils unique, and recently discovered, microbiomes on Mt Hood that only exist at glacier-fed





springs (Miller et al., 2021). As such, glacier-dependent ecosystems on Mt. Hood may be more at risk than previously surmised.

## 5 Conclusions

We find that the overall recession of glaciers on Mt. Hood in the first two decades of the 21st century is unprecedented with respect to the last 120 years, surpassing in sum the glacier retreat that occurred in the early 1900s. This century-scale retreat of glaciers can be attributed to a warming climate, with regional 30-year average temperature now ~1.1°C warmer than the 1895-1924 average. On Mt. Hood, two glaciers have ceased to be glaciers (Glisan, Palmer) with another two glaciers nearing this reclassification (Zigzag, Coalman) and a third rapidly losing area towards this threshold (Ladd). As such, the warming

climate of the last 120 years, especially the last 20 years, has significantly impacted the glacier coverage of Mt. Hood and is threatening water resources and dependent ecosystems and economies. This recession will continue unless and until the sign of multi-decadal temperature change is reversed.

## Data availability

The data referred to in this paper have all been provided within the tables and figures in the main text.

## Author contributions

The author contributions, following the CRediT authorship guidelines, are as follows: NB-F, SJB, and AEC: conceptualization; NB-F, SJB, WCS, and AEC: methodology; NB-F, SJB, WCS, and AEC: validation; NB-F, SJB, WCS, MT, and AEC: analysis; NB-F, SJB, WCS, MT, DHR, and AEC: investigation; NB-F, SJB, WCS, DHR, and AEC: resources; NB-F, SJB, WCS, MT, and AEC: data curation; AEC: original draft; NB-F, SJB, WCS, MT, DHR, and AEC:

review and editing; NB-F, SJB, WCS, MT, DHR, and AEC: visualization; NB-F, SJB, WCS, MT, DHR, and AEC: administration; NB-F, SJB, WCS, MT, DHR, and AEC: funding acquisition.

## Competing interests

The authors declare that they have no conflict of interest.



**Acknowledgements**

We thank the U.S. National Forest Service for permitting research access to the Mt. Hood Wilderness Area and Andrew
Fountain for sharing the historical area record for Mt. Hood. This work was supported by a conservation grant from the
Mazamas Mountaineering Club. The Oregon Glaciers Institute is a non-profit 501-c-3 that is supported by private donations.

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
