# Peer review of "Unprecedented Twenty-First Century Glacier Loss on Mt. Hood, Oregon, U.S.A."

_EGUsphere, 2024_

## Referee Comment (RC1)

Comments to: Unprecedented Twenty-First Century Glacier Loss on Mt. Hood, Oregon, U.S.A.

General feedback:
The manuscript provides a novel dataset on glacier outlines on Mt. Hood. Area and length changes provide valuable insights into the glacier dynamics and reaction to climate change. The text is well written but I did not focus on linguistic or grammatical errors in my review.
I would suggest some restructuring of the Methods and Results sections. Also, the creation of Data and Study region sections would be recommended. The results should focus on the area change rates (between the time periods) and the change in elevation of the terminus. Changes of topographic parameters as a proxy for glacier mass balance would give the data more weight and a better insight into climatic changes.

Abstract:
L9-19 "the 21st century behaviour of glaciers on Mt. Hood has not been directly documented". This is a bold statement since global studies like Hugonnet et al. 2021 (https://doi.org/10.1038/s41586-021-03436-z) document studies also on Mt. Hood. These elevation changes, clipped to your and 2004 outlines would give you elevation as well as volume change rates as for the glaciers.

L15 Also mention in brackets the long-term change rates (% $a^{-1}$).
L20 The study from Hugonnet does not analyse retreat but glacier elevation change.
L25-30 I would not list the number of glaciers that disappeared since this largely depends on the glacier sample. I would rather cite relative area changes of the specific regions.

Figure 1: Three inlets are not necessary. One should be enough to locate the study region. Also, satellite images as background might increase visibility.

Figure 2: Name (a) and (b). Possibly Figures 1 and 2 can be merged into one since they are showing similar data. Also, add to the figure caption that in (a) the glacier coverage of the topographic map is from the 50s.

L40: Why does Fig.1 show that data on the recent behaviour is limited? I would say Fig. 1 shows that there is data since 1907.
L61 importance not import

Introduction:
Include also volcanic history and shorten the paragraph about water and climate.

Methods
Restructure the methods section, rename it to data and methods create subsections:
1. GPS frontal positions and length changes
2. Repeat photographs (describe what you want to read out of these images)
3. Glacier inventories and mapping (list used satellite images and outline datasets)
4. Uncertainties (refer to more literature. E.g. 10.3189/2013AoG63A296)
I would also recommend to check high resolution satellite images (ESRI imagery base map) to verify your outlines. Also write clrear if you only consider debris free ice or also debris covered parts and make it consistent for all glaciers.

L80: Volcanic Activity to Introduction
L88: How did you map the glaciers, manual delineation? Please explain the approach in more detail.
How did you calculate the length changes? Did you create flowlines and if so how?

L129: How have the differing glacier response times be considered?

Results:
General comments
1. I would restructure the results section. List first results regarding the entire Mt Hood glaciers (absolute and relative area changes, also area change rates (per year)) and change of topographic parameters. Then describe the individual glaciers.
2. Also, parts of the results section are very descriptive. You could also create a study region section where you describe each individual glacier. In the results section I would focus on the changes.
3. For the change assessment, you should calculate the change of mid-point elevation = (Min+Max)/2 this can used as a change of ELA and compared to Temperature changes. Also, check how to calculate mass balance estimates from length changes. Check literature (10.3189/s0260305500015834, 10.1016/S0921-8181(02)00223-0)
4. For all glaciers, include 1907 areas and list changes since then. For all glaciers, list relative area changes, like this they are better comparable.
5. It is not clear to me what you did with the repeat photographs. Please describe the results of these photographs.

Table 1: Please also include changes in minimum elevation.
    Accum. Zone Slope: How did you separate the accumulation zone? If not described list the general slope. Include all area values into the table or create a separate table for these. Also, the headwall height might not be that relevant since it should not change substantially over time.

L141: Part about volcanic age to methods section
Fig.3-6 possibly put to supplement except for one or two examples. If in the paper than highlight what you can see in the image with annotations. Otherwise it is hard to interpret the images.
L174: Flow not flour
L291: Give relative area changes
L293-298: This can be written a bit more condensed. For example: The glacier had an area of 9.98 km$^2$ in 1907, shrunk to 7.11 km$^2$ in 2000 (-30%) and 4.3 km$^2$ in 2023 (-60%). Also include area change rates per year
Fig. 8: As for Fig. 7: Do not show Figures before they are cited in the Text. This is one of your main findings. Highlight this and describe area changes more in the text (and Table 1).

L317: Accumulation area vs accumulation zone. Check consistency
3.3 Decadal rates: I would split the study period into two parts: 1907-2000 and 2000-2023. Then calculate relative area change rates (per decade but also possibly per year). Store these values in a table. Figure 10 complements these findings very well. Absolute changes are largely dependent on the glacier's size.

Fig. 10 Not sure what the grey area indicates. Is it the uncertainty? If so, why is it so high after 2000? Please explain in the figure caption. Also, absolute area changes are not comparable as the change depends on the area.

3.4 Glacier- climate relationship
Discuss glacier response time, since this has a large influence on the correlation to the climate. Since the glaciers on Mt. Hood are fairly steep, and small and have an oceanic climate, the response time might not be very long, but it is noteworthy nevertheless. Correlating length changes directly with temperature/precipitation is risky. The approach should be better defended. Also, what is the relationship between area and temperature?

You could calculate temperature changes from ELA changes (mid-point elevation change) and compare these with the temperature observations.

L367: Include references to figures in the text and do not list all figures and tables before the main chapter.

Fig.11: instead of length changes show relative area changes.

Discussion
I am missing a section where you compare relative area change rates with glaciers from other regions. The glaciers you compare them to should also be roughly the same size.

L452: Also, what outlines/parameters the models used might have a large influence. Check.
L470: Would rename melt water to excess meltwater contribution (or imbalance contribution). It does not describe the meltwater which includes the seasonal meltwater. But in general, discussing peak water seems difficult from your data since you don't have runoff or mass change measurements. You could include data from Hugonnet et al. 2021 to calculate the volume change.
Conclusion
List main findings: area changes, length changes, and increase in change rates (with values).

---

## Referee Comment (RC2)

[referee-annotated manuscript omitted]

---

## Community Comment (CC1)

**General comments**

While I appreciate the dedication of the authors to provide full details about the observed changes of each investigated glacier on Mt. Hood (richly illustrated with before-after images from the field), the study has unfortunately taken a decision that makes the results section largely obsolete: In L95 it is written that 'debris-covered stagnant ice was separated from debris-covered active ice'. Apart from the difficulties in deciding about stagnant vs. active ice (have flow velocities been used for this?), this is not allowed for change assessment. The ice is still there and the larger glaciers flowing to the north are well connected to their upper ice body. The glacier termini are still close to the orange circles in Fig. 2B (please add A, B, C, etc. to *all* panels) and frontal retreat is minimal. Even on the low-resolution Sentinel-2 image one can see that the terminus for Coe and Elliot is about where the orange circle is. For Ladd Glacier the terminus is actually above the icon for the camera named *6A*. Indeed, the ice is highly debris covered and might not move much, but it is a part of the glacier,

Also for the area changes the authors have disregarded debris-covered ice under debris cover and found area changes that are much too high. For example, the central part of Zigzag Glacier is only covered by a large rock avalanche and still there. The northern part of the ice has been included but the central part under the debris and the southern part (to the right of the blue circle) is not considered because of stagnation (L168: 'no actively maintained crevasses'). But this is not the definition of a glacier. Of course, when the ice is disconnected and no longer fed by ice avalanches we have dead ice that requires special attention. This has to be elaborated and considered for a study with a focus on area and length changes. But here the 'unprecedented' glacier loss is due to a change of the rules rather than real changes and the obtained changes have thus no meaning.

As a note, also Glisan is not gone but very small and still there. The size limit that is usually used for glaciers is 0.01 km$^2$ rather than 0.1 km$^2$. For the classification of very small glaciers please have a loo the study by Leigh et al. (doi.org/10.1017/jog.2019.50). If a glacier fragmented over time in several parts, all parts have to be summed up for change assessment, independent of their status (e.g. being dead ice).

There are some further issues with the study such as the relation of length and area changes to climate data from the same period (which is not possible as glaciers have a response time) or the missing annotation and outlines on most of the field images (so that the reader can not see what is what) but this can be adjusted by removing the climatic interpretation and adding proper annotation to each picture. What is more a problem, is the qualitative nature of the images and the missing visualization of the applied delineation at a proper scale. For such small (and often debris-covered) glaciers as in this region the green lines on Fig. 2 B tell me nothing. This is much too coarse to properly follow what has been decided. One can certainly show clearer close-ups from the Sentinel-2 image, but for this region we have a very high-resolution image available from the ESRI Basemap that was acquired just a few days before the Sentinel-2 image. Please use it for a more sound assessment.

As a first step, I would save the available image as a geotif files at different spatial resolutions (i.e. in many pieces) so that it can also be used when the image is no longer available in the Basemap. Then redo the digitization at a high magnification and consider ice under debris cover as a part of the glacier. In the main paper you can then have a couple of these very high-resolution images with outline overlays for each of the glaciers and a detailed description of

the interpretation, in particular for the critical parts (remaining glaciers could be shown in a supplement). The photos from the field might be used to support the information, but they need to be properly annotated to follow the interpretation. Still, there could be an analysis of the changes according to certain criteria (as in Figs. 7, 8 and 10), but be aware that the results will be different. As a further note here, please omit all change rates in km$^2$ (Section 3.3 and Fig. 9) as they scale with glaciers size and are incomparable across glaciers and please provide relative area change rates in % per year rather than per decade.

Finally, please stick to what has been investigated. Peak water can be discussed when glacier mass balance and future volume change has been the topic, but this study is about past area and length changes. Please use the opportunity and elaborate on what we can do these days with freely available and geocoded very high-resolution imagery (when they are acquired at the right point in time) and show us where the problems are in such a region (with frequent rock fall) to get glacier maps at the highest possible quality. Of course, the change assessment is also very welcome but what would likely be even more beneficial is to present and discuss limits and possibilities of such images.

**Specific comments**
L29, 40, 45, etc.: For contemporary glaciers, please use glacierized instead of glaciated.
All Figures: Please provide A, B, C, etc. to all images in a panel and not just to the left ones or a few of them (as in Fig. 4). Please annotate all photos from the field so that the readers know what can be seen.

L20: The Hugonnet study is not about glacier retreat (change in length) but glacier volume and mass change.
L23: There is only a Pelto study from 2022. Maybe also cite the WGMS database (it has a doi) here, as this is likely the original source of the data.
L25: The glaciers will likely not disappear but melt away. Apart from that, more important than the number of glaciers is likely the volume loss. Please report this
L35: Figure 2 might serve as an overview for the study region, but it is way too coarse to illustrate what has been delineated (and what not)
L61: of importance
L95: This is the step that is responsible for my recommendation for rejection of the study (see general comments).
L129: One cannot compare length or area changes to climate data (T, P) of the same period as glaciers have a response time. It might be short in this region but it is not zero. Glacier length changes are a delayed, filtered and enhanced signal that might reflect climatic conditions from a very different and averaged period. A direct comparison is only possible for mass balance. Please remove the climatic interpretation,
L134: What is a 'prior-30-year average'

L166 (Zigzag): See general comments, the glacier is still there. Please note, as images with elevation contour lines or a readable scale are not shown, there is no possibility to follow the numbers given here (and for all other glaciers). This has to be changed in the revised version for all glaciers.
L174: effluent glacier flow
L183: The cited paper does not describe the separation of Zigzag. Indeed, a close view at the image of the ESRI Basemap reveals that both might still be connected. The ice is under debris cover and likely very thin, but there is a connection.
L192: None of these elevation changes can be seen on the photos.

L196: 'Remnant ice' => Please mark it on the image.

L213: Where is this landslide? Mark it on the image

L225: From which source do you know that the ice is stagnant, are there any velocity measurements?

L233: But they still had to be counted as a part of the total glacier area.

L243: Also Coe's (real) terminus is 100% debris covered.

L246: A small part of it seems still to be connected.

L259: Please include the (stagnant) ice under debris cover to determine area changes.

L267: No, the glacier is also visible in Fig. 6B (right) from 2023.

L277: What does 'communicated' mean?

L280: Instead of 'blanketing' I would write 'covering'.

L293: Why is 'Seven glaciers' bold? Please provide change rates in a table.

L301-306: Please do not show figures before they are cited in the text. Fig. 8: As mentioned above the acceleration for some of the glaciers is due to a change of the rules applied to interpret the images. Please redo the mapping and adjust the numbers to their correct values.

L322: See above, the strong retreat of Coe and Eliot is due to the missing consideration of their tongues.

L337: Please give change rates in per cent and per year as in other studies. Remove the changes in km$^2$, they cannot be compared across glaciers.

L366: Please remove Section 3.4. It makes no sense for area and length changes without considering response times.

L395: The discussion Section will likely change very much once the correct numbers are available.

L409: This is not a proper publication to define the minimum size of glaciers. Please use official ones.

L414: Why should Coalman shrink further? The glacier is fully in the accumulation area.

L496: I am not sure if this threat is real. At least it cannot be derived from the length and area changes analysed here.

L499: The maps are not sufficient. Please provide a shape file of your 2023 outlines to show where the interpretation is wrong. The figures are way too small to see the relevant details.

L513ff: I think all Rsch. should be Res.

L535: Please cite here the paper in the Annals of Glaciology

L545: cite as doi.org/jog.2023.86

---

## Author Response (AR1)

*We appreciate the comments and suggestions from the reviewers (in* unitalicized *text below). In almost all cases, we have made the suggested revisions and changes, which we outline below (in italicized text). In the few cases where we felt the suggested changes were beyond the scope of our study, we provide our reasoning below.*

**REVIEWER 1**

General feedback:
The manuscript provides a novel dataset on glacier outlines on Mt. Hood. Area and length changes provide valuable insights into the glacier dynamics and reaction to climate change. The text is well written but I did not focus on linguistic or grammatical errors in my review. I would suggest some restructuring of the Methods and Results sections. Also, the creation of Data and Study region sections would be recommended. The results should focus on the area change rates (between the time periods) and the change in elevation of the terminus. Changes of topographic parameters as a proxy for glacier mass balance would give the data more weight and a better insight into climatic changes.

*We have restructured the methods and results section as well as added more information on the study region to the introduction in terms of volcanic history. In terms of data, we have created a supplement that contains our descriptions of the glacier changes and the repeat photographs. We have focused our results on the rates of area change and terminus elevation change, but feel inclusion on mass balance proxy estimates goes beyond our existing study, mainly due to the lack of data on mass balance gradients for Oregon glaciers.*

Abstract:
L9-19 "the 21st century behaviour of glaciers on Mt. Hood has not been directly documented". This is a bold statement since global studies like Hugonnet et al. 2021 (https://doi.org/10.1038/s41586-021-03436-z) document studies also on Mt. Hood. These elevation changes, clipped to your and 2004 outlines would give you elevation as well as volume change rates as for the glaciers.

*We meant by "not been directly documented" as from direct field measurements. We have clarified this in our revised manuscript by saying "not been directly documented at the ground level."*

L15 Also mention in brackets the long-term change rates (% a-1).

*We have added in the relative rates of change but use year (yr) instead of annum (a) in keeping with Hugonnet et al. (2021).*

L20 The study from Hugonnet does not analyse retreat but glacier elevation change.

*We have added "loss of mass and retreat" to correct the sentence.*

L25-30 I would not list the number of glaciers that disappeared since this largely depends on the glacier sample. I would rather cite relative area changes of the specific regions.

*We have added the relative area information requested by the reviewer for these regions. However, we would also prefer to keep the number of glaciers that disappeared as this represents a fundamental change in the hydrology of that basin, which is lost when the summed changes of glaciers are presented. Also, we are referring to the results of Rounce et al. (2023) in terms of individual glacier disappearance. We have added reference to Robel et al. (2024), which discusses the importance of linking glaciological research to communities that depend on such research.*

Figure 1: Three inlets are not necessary. One should be enough to locate the study region. Also, satellite images as background might increase visibility.

*We have reduced the number of insets to one and added another panel to show the region and glacier records referred to in the text.*

Figure 2: Name (a) and (b). Possibly Figures 1 and 2 can be merged into one since they are showing similar data. Also, add to the figure caption that in (a) the glacier coverage of the topographic map is from the 50s.

*We have moved Figure 2 to the supplement and added an (A) and a (B) as well as indicating the glacier coverage date in (A). We have added the select photo locations that are still included in the main text to Figure 1.*

L40: Why does Fig.1 show that data on the recent behaviour is limited? I would say Fig. 1 shows that there is data since 1907.

*We have changed "recent" to "21st century".*

L61 importance not import

*We have made this change.*

Introduction:
Include also volcanic history and shorten the paragraph about water and climate.

*We have added a summary of Mt. Hood's volcanic history and changed the paragraph on water. We now more fully lay out who depends on Mt. Hood's glacier resources in keeping with Robel et al. (2024). We prefer to keep our discussion of climate impacts on Mt. Hood's glaciers unchanged because we feel this is important to highlight prior research on Mt. Hood's glaciers, and findings to date conflict with nearby glacier changes and broader scale studies of glacier change.*

Methods
Restructure the methods section, rename it to data and methods create subsections:
1. GPS frontal positions and length changes
2. Repeat photographs (describe what you want to read out of these images)

3. Glacier inventories and mapping (list used satellite images and outline datasets)
4. Uncertainties (refer to more literature. E.g. 10.3189/2013AoG63A296)

*We have restructured our Data and Methods into four sections:*
*2.1 2003 Frontal GPS Positions and Repeat Photography*
*2.2 2023 Glacier Inventory, Mapping and Uncertainty*
*2.3 Glacier Area Change and Uncertainty*
*2.4 Change in Glacier Length and Climate*

*We have also added more information on our sources of uncertainty, including the reference provided by the reviewer.*

I would also recommend to check high resolution satellite images (ESRI imagery base map) to verify your outlines. Also write clrear if you only consider debris free ice or also debris covered parts and make it consistent for all glaciers.

*We have looked at the ESRI satellite images and these date to earlier in the summer when snow cover is still substantial on many glaciers. Since our study focuses on field measured glacier margins that are supported by the most recent satellite images (i.e., weekly) during a period of rapid glacier retreat, we feel our approach provides the most accurate and up-to-date documentation of glacier area on Mt. Hood. However, per this reviewer and the community comment, we now use the ESRI basemap as the base image for our revised Figure 1.*

*We have clarified that we mapped the limits of actively flowing glacier ice both debris free and debris covered.*

L80: Volcanic Activity to Introduction

*We have moved this text to the introduction and expanded the volcanic discussion.*

L88: How did you map the glaciers, manual delineation? Please explain the approach in more detail.

*We have added more detail on our iterative mapping approach. Specifically, we manually mapped glacier extent from Google Earth and Sentinel-2 imagery in the application CalTopo. We then used CalTopo on a GPS enabled tablet to physically map/walk the glacier margins in the field, modifying the remote-sensed margins to their field-based location. We did this iteratively over four years with three seasons of field mapping to come to our September 2023 presented margins here.*

How did you calculate the length changes? Did you create flowlines and if so how?

*We have added more information on our methodology: "Specifically, we calculate the distance between the lowest observed actively flowing ice in 2023 (Fig. 1B) and the lowest actively flowing ice in 2000 of Lillquist and Walker (2006) along the same flow line they used back to 1901, which was down the center of the glacier. In the case of Newton-Clark Glacier, the length*

*was measured in the northern, due-east-facing Newton drainage (Fig. 1B). Changes in glacier length were normalized to the 1901 length of the glacier to account for different glacier sizes. Lillquist and Walker (2006) did not provide uncertainties in their glacier lengths and reported change at 1 m accuracy. Here, we report our length changes at 10 m accuracy."*

L129: How have the differing glacier response times be considered?

*We have added this information into our methods. One reason we chose a 30-year average temperature and precipitation to compare against, which we were remiss in not indicating, is that this is the calculated response time of 20-30 years for Pacific Northwest glaciers just to the north of Oregon in Washington by Pelto & Hedlund (2001). We have added this information to the methods as well as testing a 15-year average temperature and precipitation to span the 20-30 year range in response time and account for potential response times down to 11 years determined by Pelto (2016) for Mt. Baker glaciers in Washington.*

Results:
General comments
1. I would restructure the results section. List first results regarding the entire Mt Hood glaciers (absolute and relative area changes, also area change rates (per year)) and change of topographic parameters. Then describe the individual glaciers.

*We have made this change to the results, noting that we have removed the topographic parameters from our paper due to other reviewers' comments.*

2. Also, parts of the results section are very descriptive. You could also create a study region section where you describe each individual glacier. In the results section I would focus on the changes.

*This is a good suggestion. We have moved the descriptive part of our results into the supplement.*

3. For the change assessment, you should calculate the change of mid-point elevation =(Min+Max)/2 this can used as a change of ELA and compared to Temperature changes. Also, check how to calculate mass balance estimates from length changes. Check literature (10.3189/s0260305500015834, 10.1016/S0921-8181(02)00223-0)

*This is a good suggestion for a follow up study, but beyond the scope of our current study. Namely, the suggested change in mid-point elevation will be half the change in minimum elevation between 2003 and 2023. To take this further, per the provided references of Haeberli & Hoelzle (1995) and Hoelzle et al. (2023), requires data on mass balance gradients for Mt. Hood's glaciers. Such data are lacking for glaciers in the Oregon Cascades. We would be very excited to conduct such an analysis once a mass balance gradient is available for Oregon glaciers or if such gradients determined in Washington can be shown as applicable to Oregon.*

4. For all glaciers, include 1907 areas and list changes since then. For all glaciers, list relative area changes, like this they are better comparable.

*We have added this information to our new Table 2, noting such data are only available for seven glaciers. We have added relative changes from 1981 and 2015/16 for all glaciers in new Table 1.*

5. It is not clear to me what you did with the repeat photographs. Please describe the results of these photographs.

*The purpose of the repeat photographs is to show the observable change in the glaciers over the last 20 years, a period of which the remote sensed change in glacier height has indicated no significant change for Mt. Hood glaciers (i.e., Menounos et al., 2019). In our revised manuscript, we have moved these repeat photographs to the supplement. The photo sets that remain in the main text have clear demarcations of what is to be observed, as requested. We also include such demarcations in the supplement photos.*

Table 1: Please also include changes in minimum elevation.

*We have added this to Table 1.*

Accum. Zone Slope: How did you separate the accumulation zone? If not described list the general slope. Include all area values into the table or create a separate table for these. Also, the headwall height might not be that relevant since it should not change substantially over time.

*Based on input from other reviewers, we have removed both the slope and headwall elevations from the tables.*

L141: Part about volcanic age to methods section

*We have removed the discussion of volcanic age from the paper following other reviewer suggestions, but these are still in the descriptive supplement.*

Fig.3-6 possibly put to supplement except for one or two examples. If in the paper than highlight what you can see in the image with annotations. Otherwise it is hard to interpret the images.

*We have moved these figures along with the detailed glacier description into the supplement. We have added one figure to the manuscript that points out key changes for representative glaciers as suggested by the reviewer.*

L174: Flow not flour

*We meant the term "glacial flour" to describe the snowmelt stream as not having a suspended sediment load of fine particles from glacial abrasion. We now say: "or evidence of fine glacial sediment suspended in the meltwater stream."*

L291: Give relative area changes

*We have added relative glacier changes between 2015 or 2016 and 2023 for each glacier.*

L293-298: This can be written a bit more condensed. For example: The glacier had an area of 9.98 km2 in 1907, shrunk to 7.11 km2 in 2000 (-30%) and 4.3 km2 in 2023 (-60%). Also include area change rates per year

*We have reworded this section almost completely. In so doing, we have made these changes, but note that per other suggestions we now only discuss % changes in area, not square kilometers.*

Fig. 8: As for Fig. 7: Do not show Figures before they are cited in the Text. This is one of your main findings. Highlight this and describe area changes more in the text (and Table 1).

*We have moved Figure 8 forward in the manuscript as suggested and it is now Figure 4. This figure now has a separate Table 2 as well. We note that the figure placement in the submitted draft was partly dictated by the automatic formatting in the template. We now have all figures following their first mention in the text, which is easier to accommodate with the fewer number of figures.*

L317: Accumulation area vs accumulation zone. Check consistency

*We have checked and made sure that all accumulation and ablation areas as consistently termed "areas".*

3.3 Decadal rates: I would split the study period into two parts: 1907-2000 and 2000-2023. Then calculate relative area change rates (per decade but also possibly per year). Store these values in a table. Figure 10 complements these findings very well. Absolute changes are largely dependent on the glacier's size.

*We have added these data to a new Table 2.*

Fig. 10 Not sure what the grey area indicates. Is it the uncertainty? If so, why is it so high after 2000? Please explain in the figure caption. Also, absolute area changes are not comparable as the change depends on the area.

*We were remiss in not stating the gray areas are the uncertainty. We have added this into the figure caption. The reason it is high is because the uncertainty is a percent and the rate of change is also a percent and so as this rate goes up, so does the uncertainty (e.g., 3% of 30% is much greater than 3% of 3%).*

3.4 Glacier- climate relationship
Discuss glacier response time, since this has a large influence on the correlation to the climate. Since the glaciers on Mt. Hood are fairly steep, and small and have an oceanic climate, the response time might not be very long, but it is noteworthy nevertheless. Correlating length changes directly with temperature/precipitation is risky. The approach should be better defended. Also, what is the relationship between area and temperature? You could calculate temperature changes from ELA changes (mid-point elevation change) and compare these with the temperature observations.

*We have added in a discussion of glacier response time and note that our use of the backward-running 30-year mean of temperature and precipitation reflects this response time, with our additional 15-year backward running mean spanning the range in response time. Without further reference, we do not understand how correlating length with climate variables is risky. In fact, we find widespread use of glacier length as the metric to compare against climate (e.g., Harper, 1993; Haeberli and Hoelzle, 1995; Pelto and Hedlund, 2001; Oerlemans, 2005; Lillquist and Walker, 2006; Leclercq and Oerlemans, 2012; Leclercq et al., 2014; Pelto, 2016; Roe et al., 2016, 2021; Huston et al., 2021; Hoelzle et al., 2023). Given that other reviewers have not taken issue with the use of glacier length as the metric to compare against climate and its widespread practice, we have kept this part of our study unchanged but have significantly changed the surrounding manuscript as the reviewer has suggested.*

L367: Include references to figures in the text and do not list all figures and tables before the main chapter.

*We have made these changes.*

Fig.11: instead of length changes show relative area changes.

*Given the long history of relating glacier length changes with climate change (e.g., Harper, 1993; Haeberli and Hoelzle, 1995; Pelto and Hedlund, 2001; Oerlemans, 2005; Lillquist and Walker, 2006; Leclercq and Oerlemans, 2012; Leclercq et al., 2014; Pelto, 2016; Roe et al., 2016, 2021; Huston et al., 2021; Hoelzle et al., 2023), we have kept our length-climate analysis instead of conducting an area-climate analysis. This is fitting with other reviewer comments that had suggestions on how to improve the length-climate analysis, but other reviewers did not put forward the use of area. In part, this is because area has an additional factor of hypsometry. In the case of Mt. Hood, this is quite pronounced, e.g., with Newton-Clark Glacier being much wider than the adjacent narrow White River Glacier.*

Discussion
I am missing a section where you compare relative area change rates with glaciers from other regions. The glaciers you compare them to should also be roughly the same size.

*This is a good idea. We have added such a section to the discussion.*

L452: Also, what outlines/parameters the models used might have a large influence. Check.

*Based on input from another reviewer that is a co-author on the Frans et al. (2018) study (A. Fountain), we have modified this sentence to point out that these climate-related weather phenomena had yet to occur when Frans et al. were designing and then conducting their simulations. The study performed numerous model assessments against historical (up to 2011) observations. Based on our reading of the study, we do not see any issues with its design or parameters. Rather, in the period after the study was submitted to a journal (27 March 2015), significant climate-related weather events impacted Oregon including a snow drought followed by a hot summer (2015), a massive fire season (2020), and a record setting heatwave (2021) that*

*also exacerbated fires. Such events were absent from the temperature and precipitation forcing of the model that came from downscaled CMIP5 simulations of RCP4.5 and RCP8.5.*

L470: Would rename melt water to excess meltwater contribution (or imbalance contribution). It does not describe the meltwater which includes the seasonal meltwater. But in general, discussing peak water seems difficult from your data since you don't have runoff or mass change measurements. You could include data from Hugonnet et al. 2021 to calculate the volume change.

*Based on comments from other reviewers, we have removed this paragraph about peak excess meltwater.*

Conclusion
List main findings: area changes, length changes, and increase in change rates (with values).

*We have made these changes.*

**REVIEWER 2 (Andrew Fountain)**

Bakken-French et al., summarize the glacier change on Mt. Hood, Oregon over the past century with an emphasis on the past 20 years. They compiled previous efforts and updated the current glacier extent finding, like everywhere else, the glaciers are receding. The paper is important for local/regional understanding of the glacier changes that have occurred so far this century. While I don't think there is anything wrong with the data collection or the conclusions drawn from the analysis, significant problems remain.

*We are happy to hear the reviewer finds our work of import and does not take issue with any of our conclusions. Below, we adopt all of the reviewer's suggested revisions.*

The data collection methods are vague and problems encountered unknown. For example, the margins of the glaciers were mapped using a ground-level GPS and with satellite imagery. How much of the margin was mapped using each? A qualitative answer is fine. Probably not much at the highest, most difficult, and dangerous areas mapped with imagery.

*We have provided this information in the revised manuscript.*

Data sources are not well identified. For example, which Sentinel imagery. The data source for air temperature and precipitation was not exactly identified, was it instrument sourced or reanalysis? Like the Sentinel imagery, no source website was cited and no listing of data sets in an appendix. The uncertainty discussion was arbitrary. A 'conservative estimate' of 7% was applied. Why? Why not 10% or 5%. Any ground-truthing of uncertainty? The authors state that uncertainty was propagated through the time series, but no explanation or equation was provided. Finally, although the authors state they distinguished stagnant debris-covered ice from active ice, no methods were identified. Which brings into question the nature of active glaciers grading into

stagnant dead ice. How do we characterize glacier area and length changes under these conditions? This is an important question for glacier change studies, yet it was ignored here.

*In our revised dataset, we provide the date and source of the Sentinel-2 imagery we used and the means in which the imagery were accessed. We did indicate the source of the temperature and precipitation data including a website link, but these were in the reference list. We now provide them in the text as well as in the reference. We did discuss our uncertainty and its ground-truthing, finding it to be basically too small to measure. As such, we took a conservative approach and used the 2004 average uncertainty for glaciers on Mt. Hood from Jackson and Fountain (2007). We now discuss this uncertainty further and why we took a conservative approach. Lastly, we were remiss in not providing details on how we differentiated active glacial ice from stagnant glacial ice. We now list our methods that are taken from prior studies on Mt. Hood and small glacier delineation with satellite imagery.*

The structure of the paper has to be revised to be more careful to distinguish between methods and results. Some methods were found in results, and not enough methods were included to explain how some of the results were achieved. The exhaustive descriptive detail of each glacier needs to be moved into an appendix as it is not necessary in the main body of the manuscript. Also, the authors tended to summarize the contents of the tables and figures before engaging with the topic. This is very inefficient. The observation or fact should be stated followed by a citation to the table or figure.

*We have significantly revised the structure of our manuscript as suggested by this and other reviewers. The detailed glacier description is now in a supplement. We also adopt the efficient means of presenting results and discussion that the reviewer suggests.*

I was puzzled by several of the results. First, Figure 7 shows that greater the changing length of a glacier, greater the elevation change of the terminus. Isn't this to be expected for a glacier on a sloping surface? Is there something else implied here? In Table 2, the p-value of significant varied across the table. There should be a set confidence limit unless otherwise justified. In figures 11 and 12 the delta temperature and precipitation were plotted. I assume delta means difference, but relative to what?

*With regards to Figure 7 (now Figure 2), the reviewer is correct that the data presented are co-dependent. As such, we have removed the length discussion and only focus on the change in elevation—the purpose of which was to show that these glaciers have significantly changed in the last 20 years, converse to the findings based solely on geodetic methods.*

*In terms of Table 2 (now Table 3), we have clarified that the p-value presented is the actual value determined for the regression, not the significance threshold. The limit is always <0.05 for significance. We have changed the Table caption to clarify this issue.*

*For Figure 11 (now Figure 6), we now clarify that delta is change in temperature and that it is relative to the 1895-1924 average. We have removed Figure 12 as these results are better encapsulated in Table 3 with further analyses added per reviewer comments. We have also added this information in our methods, which we were remiss in not including.*

Fundamentally, I think the paper is sound but the problems summarized above require extensive revisions. Also, the text can be tightened considerably. Extensive detailed comments are included on the manuscript. I did not edit the Discussion or Conclusions significantly due to the problems found in the prior sections of the paper. Once they are resolved the contents of the latter parts of the report will become more apparent.

*We are pleased that the reviewer finds our paper to be sound and below address each of these comments in detail.*

**Line-by-Line Responses to Comments in Text:**
*26: We have changed this to indicate only ~0.017 km$^2$ of glacier ice remained in 2015.*

*44: We have added "U.S." to "Forest Service".*

*70: We have removed "cryosphere".*

*95: We now indicate how active from stagnant ice was determined in the field. Specifically, we now state: "We followed prior methods for differentiating between actively flowing debris covered ice from stagnant debris covered ice on Mt. Hood (e.g., Lillquist and Walker, 2006): an identifiable glacier front, a lobate form of the terminus, and a visible connection of the lobate terminus to debris-free flowing ice. This methodology is consistent with the observation of a convex shape of actively flowing glacier margins by Leonard and Fountain (2003) and Leigh et al. (2019). To these criteria, we added the presence of crevasses observed in the field as another indicator of ice flow whether under debris or in bare ice (Leigh et al., 2019)."*

*99: We attempted to determine an uncertainty from our remote sensed margins compared to our field-mapped margins for 2023, but found such good agreement that we felt such a smaller uncertainty was unrealistic. We chose the ±7% uncertainty because this was the average glacier area uncertainty Jackson & Fountain (2007) derived. We now indicate that this uncertainty is still higher than that found from tests conducted in the European Alps and Alaska (Paul et al., 2013). Nevertheless, we keep this uncertainty as its conservative nature makes our conclusions not dependent on our adopted ice-margin uncertainty. Specifically, we now state: "In comparing our manual remote-sensed ice margins with direct field observations, we found excellent agreement in terms of ice limits. As such, any uncertainty in these field-based glacier extents was difficult to quantify as the values were small. Using only remote sensing, Fountain et al. (2023) had their lowest ice-extent uncertainty for a Mt. Hood glacier at ±1%, while Jackson and Fountain (2007) reported an uncertainty for their 2000 Mt. Hood glacier areas of ±0%. Our uncertainty is most likely lower than ±1% but to be conservative we assumed a ±7% uncertainty in the glacier area for 2023. We based this ±7% uncertainty on the average area uncertainty for the 2004 extents of Mt. Hood glaciers determined by Jackson and Fountain (2007), which was ±7%. Our conservative ±7% is slightly greater than the uncertainties determined from testing the accuracy of remote sensed glacier area from the European Alps and Alaska of 2.6-5.7% (Paul et al., 2013). This means any conclusions we made from our mapped glacier area were likely not dependent on our assumed glacier area uncertainty."*

*102: We have made this suggested language change to "to update the glacier change…".*

*108-110: We have removed this sentenced as suggested.*

*110: We now explain how uncertainty was propagated: "The propagation of uncertainty was through quadrature where the squares of the individual percent uncertainties were summed, and their square root taken."*

*112: We have removed the "In the early 1900s…" from the sentence, fixing the reviewer's comment.*

*116: We have simplified this section, as the reviewer is correct that this does not address the overall issue with conical volcanoes. We now state "To aid comparison between glaciers of different sizes on Mt. Hood and to other glaciers in the Pacific Northwest, we calculated relative changes in glacier area where the change in glacier area was determined as a percent of its prior area." Per other reviewer suggestions, we now only focus on % change and as such this point has otherwise been removed from the paper.*

*118-120: While this list of references seems exhaustive, we feel it is important to include given such suggestions like reviewer 1's where they suggested to not do such assessments between glacier length and climate and rather focus on other metrics.*

*121-126: We have made these suggested changes, moving the additional detail to the results section.*

*130: We had provided a website link in the references; we now have added it to the text.*

*132-133: We were remiss in not describing our reasoning behind these monthly temperature and precipitation groupings. We now state: "For temperature, we focused on changes in annual, May-October and June-September temperature. May through October is generally the period of time in the Oregon Cascades that includes from peak winter snowpack to the first snowfall of the next winter, or the maximum length of the ablation season. Since this can vary, we also examined June-September temperature as a shorter ablation season period. Indeed, Pelto (2018) determined the ablation season to be May-September or June-September for glaciers in Washington. For precipitation, we analyzed annual, November-April and October-May precipitation. November through April in Oregon generally spans the time period from the first winter snowfall until peak spring snowpack, or the accumulation season. To account for variability in this season, we also used October through May precipitation."*

*134: This is a good suggestion on glacier response timescales. In a sense, this is what we are approximating. We have added in text on glacier response timescales, focusing on results from Pelto & Hedlund (2001) that found a response time of 20-30 years for glaciers similar to those on Mt. Hood. Thus, our 30-year backward running mean approximates a 30-year response time. To this we add a 15-year backward running mean analysis to span the 20-30 year range in glacier response time and account for response times down to 11 years noted by Pelto (2016). We have removed the point on NOAA's definition of a climate interval of time.*

*134-135: Correct, these are reanalysis data for the area of Hood River County. We now explain this as follows: "For records of regional climate change, we used U.S. National Oceanic and Atmospheric Administration (NOAA) temperature and precipitation reanalysis data for the region of Hood River County (Fig. 1) from 1895 to 2023 (U.S. NOAA, 2024, https://www.ncei.noaa.gov/access/monitoring/climate-at-a-glance/). We chose Hood River County as this is the smallest geographic region for the reanalysis data that includes these five glaciers and would still have direct weather station measurements within the geographic domain over the period of comparison."*

*136: We have added reference to the snow telemetry sites and their information on operation: "Snow telemetry stations on and around Mt. Hood were only installed in the late 1970s, with temperature sensors beginning to work consistently in the late 1980s (U.S. National Resources Conservation Service, 2024, https://www.nrcs.usda.gov/conservation-basics/conservation-by-state/oregon/oregon-snow-survey/about-us)."*

*139: Yes, they are referring to the U.S.G.S. officially named glacier data. We used the 2016 U.S. Forest Service map in Figure 2 as it is the most recent official U.S. map product that includes the U.S.G.S. officially named glaciers. In revising our paper, this sentence has been cut to conform with the general reorganization suggested by reviewer 1 (and this reviewer as well).*

*142: We have added the volcanic history into the introduction and included a link to the website directly in the text in addition to the link in the reference list.*

*142-145: We added the information on the Fountain et al. (2023) glacier area data to the methods section. We removed the other two sentences that reiterated our methods already described above in the text.*

*147-150: We have removed this paragraph as suggested by the reviewer and it is no longer relevant with the changes we have made to the manuscript.*

*Table 1: We have significantly changed Table 1 to meet the revisions suggested elsewhere in the manuscript. This has removed the headwall and accumulation zone slope columns. We note that we do not have 2003 glacier area data as that was not collected in that year. We have removed the horizontal lines from the table.*

*151-292: We have moved this section to the supplement. It has also been revised according to the reviewers' suggestions.*

*293: We have removed the bold text. We have significantly changed the results section in accord with reviewer's suggestions.*

*Figure 7: The reviewer is correct as these are two related metrics: terminus elevation change and glacier length change. In our revised manuscript, we only focus on the terminus elevation change between 2003 and 2023. This figure is changed accordingly (now Figure 2). Our length change discussion is now focused on the larger scale change in length from 1901 up to 2023.*

*Figure 8: There is uncertainty on the combined record, which we indicated in the figure caption stating, "note that the uncertainty on the combined record is smaller than the symbol size." This is now Figure 4.*

*311: Given this expected relationship, we have removed this comparison from the paper.*

*313-320: We have removed this text as suggested by the reviewer, which also addresses the comment on line 316.*

*323: Given the changes to the manuscript from this reviewer's comments and the other reviewers' comments, this comment is no longer relevant.*

*323-326: We have removed this sentence, as suggested.*

*Figure 9: We have removed this figure per comments from other reviewers.*

*367-369: We have removed this sentence.*

*Table 2: Table 2 is now Table 3. We now use and define the acronyms Ann, M-O, J-S, N-A, O-M, T and P in the table caption. We use these acronyms to define Annual, May-October, June-September, November-April, October-May, Temperature and Precipitation. We have bolded significant correlations and removed the asterisks that we discuss in the text as well. Note that we were reporting the p results and not listing them as a threshold. The threshold was a consistent <0.05. In the new Table 3, we only list the $r^2$ results and bold those with $p < 0.05$.*

*Figure 11: We have added to our methods how we calculated the "prior 30-year average", or as we now call it "30-year backward running mean", for precipitation and temperature. We now have normalized the glacier length records to their 1901 length and analyze the changes in normalized length relative to temperature and precipitation changes. We have also changed all the terminology to be "30-year backward running mean" or "15-year backward running mean". Figure 11 is now Figure 6.*

*Figure 12: We meant by delta the change in temperature or precipitation; we now indicate the delta means change. The gray line was the regression result for 1901-2000, which we indicated in the figure caption. However, we have removed the gray line from our revised paper as its results are provided in Table 3, making it redundant. We hope this will avoid further confusion.*

*430: We have added in information on glacier response time. Specifically, we now say: "Averaging three years of weather is shorter than the 11- to 30-year response time for glaciers like those on Mt. Hood (Pelto and Hedlund, 2001; Pelto, 2016), which we approximate with our 30-year and 15-year backward running means."*

*452-453: We agree and now indicate the following: "We hypothesize that the observed faster retreat of glaciers on Mt. Hood relative to their simulated retreat may, in part, be due to climate-*

*related weather phenomena that had yet to occur and thus were not included in the model forcing of Frans et al. (2016)."*

**REVIEWER 3 (Mauri Pelto)**

The authors provide a detailed report on the rapid recession of Mount Hood, Oregon glaciers in the last ~40 years. Several items must be addressed to make this a useful addition to the literature documenting the rapid demise of these glaciers. There are too few glaciers too discern the impact of geographic characteristics on rate of change, focus less time on trying to discern sensitivity in this regard. Discussion of data from Driedger and Kennard (1984) is imperative. Referencing the response time of Pacific Northwest glaciers to climate change. In the discussion or results relate Mount Hood results to those from Mount Rainer and Mount Baker.

*We appreciate that the reviewer finds our report detailed and of use (once revised) for documenting rapid demise of glaciers. We have removed the discussions of geographic characteristics and sensitivity for the different glaciers due to the small sample number. We have added further discussion of Driedger & Kennard (1986) to our study, including their ice areas to a new Table 1. And we now discuss the response time of Pacific Northwest glaciers as well as compare our results to Mt. Rainier and use the Mount Baker results to support our findings.*

18: "with attendant consequences for water resources" More specifics are warranted here, i.e.. There are PNW observations of significant declines in late summer glacier runoff contributions, which reduces baseflow and minimum flows. What are the water resources utilized for?

*We have removed this latter part of the last sentence of the abstract per comments from other reviewers.*

30: Worth noting glacier loss in the North Cascades and on Mount Shasta both north and south of study area, there are 31 glaciers in the North Cascades noted as having disappeared in last 30 years in the GLIMS extinct glacier category.

*We have added in the GLIMS extinct glacier category data here (our count is 33 glaciers are extinct). We would like to add similar information for Mt. Shasta, but to our knowledge such recent data are lacking. No glaciers are listed as extinct in GLIMS and the last assessment of glacier extent is from 2003 (Howat et al., 2007), limiting our knowledge of how these glaciers have behaved in the last 20 years.*

45: Driedger and Kennard (1986) did map all glaciers on the mountain and provided detailed maps of extent and thickness on several glaciers from 1981 field observations. They found an area of 104 million ft2. This information should be incorporated or explain why it is not.

*This is a good suggestion. We have added this information to the paper as well as Table 1 where we calculate the % change in glacier area from 1981 to 2023. We now discuss this change in our manuscript and include it in the abstract.*

55: Here or later might want to comment on likely cause of this error, that is also evident on North Cascade glaciers in that the glacier footprint used on some of the small glaciers included non-glaciated areas.

*We discuss this cause of error later on in the discussion section of both original and revised manuscript, referencing the recent Florentine et al. (2024) paper on this topic.*

92: Particularly in 2021 and 2023 large regional glacier mass balance losses were observed across the PNW, which helped better expose the margins of the glaciers.

*This is a good idea to point out, which we now do in our revised paper as well as note that 2020 had an exceptionally long summer in Oregon that resulted in a similar very low end-of-summer snow coverage on glaciers.*

106: Evaluate the ability to use the 1981 extents from (Driedger and Kennard, 1986). This is useful timing because it follows the period of expansion of many of the Cascade glaciers from 1950-1980.

*Since the Driedger & Kennard (1986) ice extents are only single year measurements, we do a separate comparison to our 2023 extents rather than include in the time series of area measurements for seven glaciers by Jackson & Fountain (2007). These data are in a revised Table 1 and are added to our results and discussion sections as well as abstract.*

159: What is the mean slope of the steep upper section, steepest part says little.

*We have removed discussion of slope and now just indicate its accumulation area is between the two rock walls.*

170: "a 30% decline in its area in the last eight years."

*Based on another reviewer's suggestion, we now indicate that "the glacier had lost about 40% of its area in the last eight years."*

243: Has the less debris cover been a consistent feature for Coe's terminus?

*Yes, at least for the period of 2003-2020/2021/2023, which we now indicate.*

250: Up to 60% not a useful metric, provide a steepness metric that is less of a point measurement.

*We have removed accumulation area slope from the paper.*

260: What does shallow reference here, slope, snow depth etc?

*We have removed discussion of slope.*

267: This fits with the rapid area loss seen on Mount Rainier that is leading to loss of glacier status. I have not found on the slopes of Cascade volcanoes that active ice exists below the 0.03 km2 threshold. Report the observed area in 2023 even if below threshold, can use Sentinel imagery for this.

*We have added in the area of this stagnant remnant glacial ice, which is ~0.009 km².*

276: Sandy Glacier is the only glacier with a recent/current ice cave system. Other glaciers such as White River have had ice caves in the past.

*We now indicate that it is the only glacier with a cave system at present.*

298: Could add that "The rapid area loss is indicative of large mass balance losses."

*We have added this statement.*

310: Can provide a quantity here as to the significance of the retreat distance.

*We have significantly revised this section and provide more context on the significance of ice margin change. However, given the major revisions we have conducted, this comment is no longer relevant.*

316: The number of glaciers is too small to discern how important each variable is, this section could be removed. The overall point is regardless of specific characteristics retreat has been significant.

*We have removed this section.*

327: Reword –"When comparing adjacent glaciers, topographic and other factors are key to driving different amounts of retreat."

*We have made this change.*

338: Figure 8 does a good job of providing a visual of the acceleration of area loss after ~2000 on five of the seven glaciers. I am not sure of the added value of breaking this up by decade in Figure 9, when measurements do not always fit nicely into decadal periods. If you keep Figure 9 a mechanism to better distinguish periods of gains from those of losses is necessary. Reflect here or in discussion on the comparison with retreats reported from Mount Rainier Figure 10 (Beason et al 2023). Could look at retreat rates of Mount Baker glaciers also (Harper, 1993; Pelto, 2016).

*We have removed Figure 9 from the revised paper and rather just show the area records in the old Figure 8 (now Figure 4) and then the percent change per year in our old Figure 10 (now Figure 5).*

340: This section is really difficult to read just because of how many different area loss rates and periods are mentioned. Let the table and figure provide the details and try to summarize more. All of the information is good, a reader just gets lost in the trees.

*We have significantly revised this section in accord with the reviewer's comment.*

380: Several studies have indicated that the strongest temperature relationship to Cascade glacier mass balance is the June-September period, with May and October some years being part of the accumulation season. It is fine to use May-October, but does June-September yield better results?

*We now use both May-October and June-September for analysis along with annual temperature. Similarly, we have added an analysis of October-May precipitation.*

384: Figure 11 displays well the significant rising trend in annual and ablation season temperature. Worth noting that is sustained over a sufficient period for significant glacier response. There is not a consistent precipitation trend like this since 1980 when area losses began. Would strengthen paper to reference glacier response time for Mount Hood glaciers.

*We have added in a discussion of glacier response times for Mt. Hood glaciers, referencing Pelto & Hedlund (2001) and Pelto (2016).*

396: This is an excellent statement so add quantitative measures for each of the three either means or greater than values.

*We have done this in our revised manuscript.*

422: Should reference the conclusion of Beason et al (2023) about Mount Rainier glacier change. "All 28 glaciers at Mount Rainier are in retreat, losing an average of -0.430 km2 per year parkwide during the last 125 years. Since 2015, this rate has increased to -0.544 km2 per year parkwide." Harper (1993) good reference for the changes in forcing and terminus response for 1900-1970's

*We include the unprecedented nature of Mt. Rainier glacier retreat in our revised paper as requested. Based on other revisions, we do not reference Harper (1993) here because we focus on the 21st century change in glaciers, which is the focus of our data collection and paper.*

426: Worth comparing your results to the figure 6 results of O'Neel et al (2019) for the benchmark glaciers.

*We have removed this section to focus our study specifically on the new data we provide about 21st century glacier change.*

452: The recent period of sustained warming has also strengthened the temperature signal. Should also note the mass balance climate relationships that drive the terminus changes, they are annually based of course, but because they are not cumulative provide good guidance for

interpreting your relationship data. Pelto (2018) notes correlations with annual balance for April 1 SWE and ablation season temperature of greater than 0.64 for all 11 North Cascade glacier noted. O'Neel et al (2019) note the lack of a trend in winter balance at benchmark glaciers, while there is a summer balance trend.

*We have added in discussion of these mass balance relationships.*

468: Nolin et al (2010) indicated peak water had been reached at Elliott Glacier-"Our SRM simulations showed that Eliot Glacier discharge increases 13% for every 1°C increase, but decreases 9% for every 10% decrease in glacier area. The recession of the Eliot Glacier in the last century already exceeds this rate; therefore, its discharge has likely been decreasing over time and will continue to decrease."

*This is a good point that we were remiss in not making. However, due to comments from other reviewers, we have removed the paragraph on peak meltwater.*

Driedger, C L, Kennard, P M 1984 Ice volumes on the Cascade volcanoes: Mount Rainier, Mount Hood, Three Sisters, and Mount Shasta. US Geological Survey. Open File Report 84–581, https://doi.org/10.3133/pp1365.

Harper JT (1993) Glacier terminus fluctuations on Mt. Baker, Washington, USA, 1940–1980, and climate variations. Arct Alp Res 25:332–340

O'Neel S, McNeil C,Sass L, Florentine C, Baker E, Peitzsch E, McGrathD, Fountain A, Fagre D (2019). Reanalysis of the US Geological Survey Benchmark Glaciers: long-term insight into climate forcing of glacier mass balance. Journal Glaciology, 65, 850–866. https://doi.org/10.1017/jog.2019.66

Pelto, M. (2016). Climate Driven Retreat of Mount Baker Glaciers and Changing Water Resources. SpringerBriefs in Climate Studies https://doi.org/10.1007/978-3-319-22605-7

*We have added these references, but note that Driedger & Kennard (1986) was already in our reference list with respect to Palmer's status as a glacier in 1981.*

**COMMUNITY COMMENT 1 (Frank Paul)**

While I appreciate the dedication of the authors to provide full details about the observed changes of each investigated glacier on Mt. Hood (richly illustrated with before-after images from the field), the study has unfortunately taken a decision that makes the results section largely obsolete: In L95 it is written that 'debris-covered stagnant ice was separated from debris-covered active ice'.

*We now fully explain that we are documenting active glacier flowing ice extent on Mt. Hood. We compare this to past area records of actively flowing glacier ice on Mt. Hood. Likewise, we*

*compare these results to changes in actively flowing glacier area on other western U.S. mountains. If we did not use these methods, our results would not be comparable to past glacier extents or changes in glaciers on other mountains. Two other reviewers who work extensively on western U.S. glaciers (A. Fountain, M. Pelto) found no issue with our approach for mapping actively flowing glaciers. Likewise, an anonymous reviewer encouraged us to compare our changes in actively flowing glacier to changes on nearby mountains where similar measurements have been made. We now clarify this in our revised manuscript.*

Apart from the difficulties in deciding about stagnant vs. active ice (have flow velocities been used for this?), this is not allowed for change assessment. The ice is still there and the larger glaciers flowing to the north are well connected to their upper ice body. The glacier termini are still close to the orange circles in Fig. 2B (please add A, B, C, etc. to *all* panels) and frontal retreat is minimal. Even on the low-resolution Sentinel-2 image one can see that the terminus for Coe and Elliot is about where the orange circle is. For Ladd Glacier the terminus is actually above the icon for the camera named *6A*. Indeed, the ice is highly debris covered and might not move much, but it is a part of the glacier,

*We have added further documentation on our methodology including how we mapped actively flowing from stagnant ice. Our study is field based and only used Sentinel-2 imagery as backup to field mapping. Our terminus outlines up into the accumulation area are all based on three years of extensive field mapping of each glacier. Consequently, our results are not derived from low-resolution imagery.*

Also for the area changes the authors have disregarded debris-covered ice under debris cover and found area changes that are much too high. For example, the central part of Zigzag Glacier is only covered by a large rock avalanche and still there. The northern part of the ice has been included but the central part under the debris and the southern part (to the right of the blue circle) is not considered because of stagnation (L168: 'no actively maintained crevasses'). But this is not the definition of a glacier. Of course, when the ice is disconnected and no longer fed by ice avalanches we have dead ice that requires special attention. This has to be elaborated and considered for a study with a focus on area and length changes. But here the 'unprecedented' glacier loss is due to a change of the rules rather than real changes and the obtained changes have thus no meaning.

*We refer the reviewer to the findings of Fountain et al. (2023) that mapped glacier extents solely from remote sensing imagery from 2015/2016. In all cases, our 2023 glacier areas are smaller than glacier areas Fountain et al. found. If we used the approach of Paul, then glaciers on Mt. Hood dramatically grew in the last 7-8 years. Again, we use a well-established methodology for documenting glacier change in the western U.S.; there is no change in rules from prior studies in this region as all studies only mapped actively flowing glacier ice as part of a given glacier. The U.S. Geological Survey defines a glacier as "a large, perennial accumulation of crystalline ice, snow, rock, sediment, and often liquid water that originates on land and moves down slope under the influence of its own weight and gravity." The Cambridge Dictionary defines a glacier as "a large mass of ice that moves slowly". In both instances, it is movement that defines a glacier, and it is this property that has been the focus of past glacier inventories and change in the western United States.*

As a note, also Glisan is not gone but very small and still there. The size limit that is usually used for glaciers is 0.01 km2 rather than 0.1 km2. For the classification of very small glaciers please have a loo the study by Leigh et al. (doi.org/10.1017/jog.2019.50). If a glacier fragmented over time in several parts, all parts have to be summed up for change assessment, independent of their status (e.g. being dead ice).

*We do not base our classification of stagnant glaciers upon its size, though this can be a helpful indicator. Rather, this is from our direct field analysis of the glacier having no evidence of flow, including the lack of crevasses that is one piece of evidence that Leigh et al. use. We now include an additional discussion of the Leigh et al. study, but note that it is for remote sensing, and we are conducting field studies supported by remote sensing.*

There are some further issues with the study such as the relation of length and area changes to climate data from the same period (which is not possible as glaciers have a response time) or the missing annotation and outlines on most of the field images (so that the reader can not see what is what) but this can be adjusted by removing the climatic interpretation and adding proper annotation to each picture. What is more a problem, is the qualitative nature of the images and the missing visualization of the applied delineation at a proper scale. For such small (and often debris-covered) glaciers as in this region the green lines on Fig. 2 B tell me nothing. This is much too coarse to properly follow what has been decided. One can certainly show clearer close-ups from the Sentinel-2 image, but for this region we have a very high- resolution image available from the ESRI Basemap that was acquired just a few days before the Sentinel-2 image. Please use it for a more sound assessment.

*We used 30-year "climate" periods precisely to reflect the response time of glaciers. We now discuss this in detail. It is exactly because of this response time that we did not use yearly changes in temperature.*

*We have added annotation to the field photographs. We also provide these repeat photographs to show that indeed glaciers have changed on Mt. Hood in the last 20 years, in direct contradiction to the geodetic results of Menounos et al. (2019) that put forward these glaciers had minimal change in the 21st century. We now better clarify the purpose of these photographs.*

*Figure 2B was used to show where 2003 measurements and photographs were taken and show our 2023 mapped ice margins. We use the Sentinel-2 imagery year after year and our mapping approach was iterative. We now clearly lay out in our revised methods our approach, which shows how our margins are field based, not solely remote sensed, and that we used rapidly acquired imagery that we then took immediately into the field to properly map glacier margins. Also, in looking at the ESRI 2023 basemap, we find that this was taken when a fair amount of snow cover still obscured ice-margin locations on Mt. Hood. This is why we chose lower resolution but more frequent Sentinal-2 imagery as we can use imagery from each year (2020 to 2023) acquired at the lowest point of snow cover as well as the image acquired right before we went into the field. We take this most recent image with us to then be modified by walking glacier margins in the field using a GPS enabled tablet.*

*However, we now include the ESRI 2023 basemap in our revised Figure 1.*

As a first step, I would save the available image as a geotif files at different spatial resolutions (i.e. in many pieces) so that it can also be used when the image is no longer available in the Basemap. Then redo the digitization at a high magnification and consider ice under debris cover as a part of the glacier. In the main paper you can then have a couple of these very high-resolution images with outline overlays for each of the glaciers and a detailed description of the interpretation, in particular for the critical parts (remaining glaciers could be shown in a supplement). The photos from the field might be used to support the information, but they need to be properly annotated to follow the interpretation. Still, there could be an analysis of the changes according to certain criteria (as in Figs. 7, 8 and 10), but be aware that the results will be different. As a further note here, please omit all change rates in km2 (Section 3.3 and Fig. 9) as they scale with glaciers size and are incomparable across glaciers and please pro- vide relative area change rates in % per year rather than per decade.

*The reviewer does not acknowledge that our mapped glacier extents are from field mapping that was supported by the most recent Sentinel imagery. If we were conducting a remote-sensing study of all snow and ice on Mt. Hood, then yes, this is a good approach. However, we are field mapping actively flowing glaciers and as such these comments above do not apply to our study. We now report the rates of glacier area change only in % per year.*

Finally, please stick to what has been investigated. Peak water can be discussed when glacier mass balance and future volume change has been the topic, but this study is about past area and length changes. Please use the opportunity and elaborate on what we can do these days with freely available and geocoded very high-resolution imagery (when they are acquired at the right point in time) and show us where the problems are in such a region (with frequent rock fall) to get glacier maps at the highest possible quality. Of course, the change assessment is also very welcome but what would likely be even more beneficial is to present and discuss limits and possibilities of such images.

*We have removed the paragraph on peak meltwater from the paper. Unfortunately, we do not understand how to reply to the rest of this paragraph, which fails to acknowledge that the point of our study was field-based mapping of actively flowing glaciers.*

**Specific comments**
L29, 40, 45, etc.: For contemporary glaciers, please use glacierized instead of glaciated

*We have made this change.*

All Figures: Please provide A, B, C, etc. to all images in a panel and not just to the left ones or a few of them (as in Fig. 4). Please annotate all photos from the field so that the readers know what can be seen.

*We have made these changes.*

L20: The Hugonnet study is not about glacier retreat (change in length) but glacier volume and mass change.

*We now state "global loss of mass and retreat of glaciers"*

L23: There is only a Pelto study from 2022. Maybe also cite the WGMS database (it has a doi) here, as this is likely the original source of the data.

*We have fixed this typo and added reference to the WGMS.*

L25: The glaciers will likely not disappear but melt away. Apart from that, more important than the number of glaciers is likely the volume loss. Please report this

*We disagree as the loss of a glacier is of importance to that watershed in addition to the global volume loss. We have added in the percent volume loss as requested.*

L35: Figure 2 might serve as an overview for the study region, but it is way too coarse to illustrate what has been delineated (and what not)

*We have moved Figure 2 into the supplement where it is used as reference for the point measurements from 2023 and repeat photograph locations. We respectfully disagree that it is too coarse to show these locations. In moving it to the supplement, we hope this corrects any confusion. Similarly, to help with potential confusion, we now use the Esri basemap in Figure 1.*

L61: of importance

*We have made this change.*

L95: This is the step that is responsible for my recommendation for rejection of the study (see general comments).

*We respectfully disagree. We now point out that we are mapping actively flowing glacier ice. We now clearly point out that this is the approach taken for mapping glacier change in the western United States, providing example studies. For instance, if we mapped stagnant ice along with flowing ice for 2023, then the glaciers on Mt. Hood would have undergone rapid expansion in area relative to the 2015/2016 extents mapped by Fountain et al. (2023). Similarly, we would not be able to compare glacier change relative to the 20th century because these records of Mt. Hood glaciers are based on the mapped extent of actively flowing glacier ice (Driedger and Kennard, 1986b; Lillquist and Walker, 2006; Jackson and Fountain, 2007). Furthermore, we would not be able to compare the rate of glacier loss on Mt. Hood to other western U.S. glaciers, as other reviewers have requested, because all of these records are from mapped actively flowing glacier ice (Fagre et al., 2017, Garwood et al., 2020; Fountain et al., 2022; Beason et al., 2023). Lastly, three other reviewers found no issue with our approach of mapping actively flowing glacier ice and rather requested more information on how we excluded stagnant ice from our mapping.*

*We hope the clarification of our approach in our revised manuscript alleviates this concern.*

L129: One cannot compare length or area changes to climate data (T, P) of the same period as glaciers have a response time. It might be short in this region but it is not zero. Glacier length changes are a delayed, filtered and enhanced signal that might reflect climatic conditions from a very different and averaged period. A direct comparison is only possible for mass balance. Please remove the climatic interpretation,

*In our revised manuscript, we address this comment. By saying climate, rather than weather, we were implying a persistent average climate state to which the glacier would adjust. Specifically, we look at climate data averaged over the prior 30 years relative to the glacier length change. We do this precisely for the reasons that the commentator raises. We now include discussion of glacier response time, which is 20-30 years for these glaciers and potentially down to 11 years. We span this range by also analyzing the average climate (temperature and precipitation) of the prior 15 years.*

L134: What is a 'prior-30-year average'

*Based on reviewer 2's comment, we realize that this is a confusing term. We now call this a "30-year (or 15-year) backward running mean".*

L166 (Zigzag): See general comments, the glacier is still there. Please note, as images with elevation contour lines or a readable scale are not shown, there is no possibility to follow the numbers given here (and for all other glaciers). This has to be changed in the revised version for all glaciers.

*As discussed above and in our manuscript, we focus on mapping only actively flowing glacier ice, which is delimitated from stagnant glacier ice. We hope our manuscript revisions makes this clear as this issue has not been raised by other reviewers.*

L174: effluent glacier flow

*This read correctly "glacial flour". To avoid confusion, we now state: "fine glacial sediment suspended in the meltwater stream."*

L183: The cited paper does not describe the separation of Zigzag. Indeed, a close view at the image of the ESRI Basemap reveals that both might still be connected. The ice is under debris cover and likely very thin, but there is a connection.

*It is true the paper only discusses the separation from White River Glacier, and we were remiss in not stating its current separation from Zigzag Glacier is based on our field-based mapping. As such, we have modified the sentence to: "Indeed, Coalman was part of the accumulation area of White River and Zigzag glaciers in the 1800s, separating from White River Glacier by 1912 (Lillquist and Walker, 2006) and later from Zigzag Glacier (unknown timing but it was separate by 2016 according to Fountain et al. (2023))."*

L192: None of these elevation changes can be seen on the photos.

*That is correct. We are relying here on our mapping of Coalman Glacier as the repeat photography was not possible in 2023 due to cloud cover. Rather, we are providing our best ability to have similar photographs from the past of Coalman Glacier that were not directly designed for repeat photography.*

L196: 'Remnant ice' => Please mark it on the image.

*We have added this label to the photographs.*

L213: Where is this landslide? Mark it on the image

*We have added this label to the photographs.*

L225: From which source do you know that the ice is stagnant, are there any velocity measurements?

*We did not have velocity measurements on this glacier in 2003. Rather, we used common field-based observations to differentiate between actively flowing glacier ice and stagnant glacier ice. We now have added these to our methods section.*

L233: But they still had to be counted as a part of the total glacier area.

*Since we are mapping and cataloguing only actively flowing glacier ice, similar to Fountain et al. (2023) and other studies, we prefer to keep these different demarcations.*

L243: Also Coe's (real) terminus is 100% debris covered.

*We have changed the wording to reflect this: "While Coe's terminus is fully covered in debris, this cover was less than on Eliot's terminus at least in 2003-2020/2021/2023."*

L246: A small part of it seems still to be connected.

*We did not state that the active ice was fully disconnected from the stagnant ice, but only that actively flowing ice was above the cliff and stagnant ice below the cliff. There is a band of stagnant ice that does extent up on the side to the region of actively flowing ice.*

L259: Please include the (stagnant) ice under debris cover to determine area changes.

*We have not made this change because this falls outside the focus and scope of our study. We are mapping actively flowing glacier ice and differentiate this from stagnant ice. In all the glacier inventories we compare our results against in the western U.S., this is the standard practice. We hope we have clarified our reasoning for our approach. If we mapped the stagnant ice with the active ice for 2023, then many of Mt. Hood's glaciers would have "grown" in area between our 2023 measurements and the 2015/2016 glacier areas of Fountain et al. (2023). We can only*

*make our comparison to past glacier dimensions on Mt. Hood and in other western U.S. glaciated ranges by mapping actively flowing glacier ice.*

L267: No, the glacier is also visible in Fig. 6B (right) from 2023.

*Unfortunately, we do not understand this comment as the stagnant glacier ice is "nearly gone", which to us means it still has visible remnant ice that is seen in Figure 6B. Note this is now Figure A9C, D.*

L277: What does 'communicated' mean?

*We have changed "communicated" to "connected".*

L280: Instead of 'blanketing' I would write 'covering'.

*We have changed "blanketing" to "covering".*

L293: Why is 'Seven glaciers' bold? Please provide change rates in a table.

*We have removed the bolding that was previously included to be similar in style to the bolding of individual glaciers in the preceding paragraphs.*

L301-306: Please do not show figures before they are cited in the text. Fig. 8: As mentioned above the acceleration for some of the glaciers is due to a change of the rules applied to interpret the images. Please redo the mapping and adjust the numbers to their correct values.

*We have fixed this layout of figures, which was partly done to accommodate the auto-formatting in the template document. All figures now follow their first mention. There is not a "change in rules" applied to the images for mapping glaciers in Figure 8. Rather, all glacier areas are only for their actively flowing components. If stagnant ice was included, then several of these glaciers with debris-covered stagnant ice regions would undergo an artificial period of growth between 2015/2016 (Fountain et al., 2023) and 2023. We have kept the figures and mapping unchanged. Note that Figure 8 is now Figure 5.*

L322: See above, the strong retreat of Coe and Eliot is due to the missing consideration of their tongues.

*Please see above. This is not due to a change in mapping standards and if we included these stagnant tongues, then these glaciers would have grown considerably in the last seven to eight years relative to the 2015/2016 areas determined by Fountain et al. (2023).*

L337: Please give change rates in per cent and per year as in other studies. Remove the changes in $km^2$, they cannot be compared across glaciers.

*We now provide the changes and rates of changes in percent and percent per year.*

L366: Please remove Section 3.4. It makes no sense for area and length changes without considering response times.

*We keep this section because we now include discussion of glacier response times. In short, because we use a 30-year or 15-year backward running mean (with the former also being in our original manuscript), we address this concern. Namely, we compare the change in glacier length relative to how much the prior 30-year (or 15-year) average temperature or precipitation has changed. This 30-year to 15-year window is chosen to reflect the 30-20 year response time of glaciers like those on Mt. Hood that could have responses times as low as 11 years. Our revised manuscript clarifies this, addressing the commentator's concern.*

L395: The discussion Section will likely change very much once the correct numbers are available.

*We have reworked our discussion section following the input of the reviewers. However, our conclusions have remained the same, which is consistent with findings of other reviewers that our study is robust.*

L409: This is not a proper publication to define the minimum size of glaciers. Please use official ones.

*The U.S. Geological Survey is an official source on glacier size within the United States. However, we now indicate that other groups consider the minimum size of a glacier to be 0.01 km$^2$ (Pfeffer et al., 2014). We have added this reference. It does not change our conclusions because we base our demarcation between actively flowing and stagnant, dead glaciers on observable field evidence, not their area.*

L414: Why should Coalman shrink further? The glacier is fully in the accumulation area.

*In the last 7 years, Coalman lost 42% of its area despite having the highest terminus on Mt. Hood. Given this rate of recent retreat, we do not see why it would all of a sudden stop despite the climate continuing to warm.*

L496: I am not sure if this threat is real. At least it cannot be derived from the length and area changes analysed here.

*We have modified the conclusions per suggestions of other reviewers, which may alleviate this concern. However, we point out that the projections for future water resources that these communities rely upon have only 1-4% decline in glacier coverage through 2060. In contrast, we find that in the last 7-8 years, these glaciers have lost 6-32% of their area (41% if Langille is considered), demonstrating that existing planning has grossly under-projected glacier loss. In as much as these ecosystems and economies depend on glacier meltwater, then they are already threatened. We have modified the language to reflect these observations.*

L499: The maps are not sufficient. Please provide a shape file of your 2023 outlines to show where the interpretation is wrong. The figures are way too small to see the relevant details.

*We will provide upon acceptance of our publication a shape file of our 2023 glacier outlines for actively flowing glacial ice that will be archived in a repository like the National Snow and Ice Data Center ([www.nsidc.org](www.nsidc.org)) that will be referenced in the final published paper.*

*Our mapping of glacier flowing ice is the standard practice for glacier mapping in the western U.S. and how past studies have measured glacier change. We now clearly state that we are assessing the change in area and length of actively flowing glaciers. Moreover, the commentator misses that our findings are based on field mapping that was only assisted by satellite imagery. In other words, it is not a satellite-based remote sensing study.*

L513ff: I think all Rsch. should be Res.

*We have made these changes.*

L535: Please cite here the paper in the Annals of Glaciology

*We have added the 1986 reference to Dreidger & Kennard in Annals of Glaciology in addition to their U.S. Geological Survey report.*

L545: cite as doi.org/jog.2023.86

*We have made this change.*

---

## Referee Report (RR1)

The revised ms has greatly improved compared to the first version. The now annotated photos guide the reader much better through the observations, the methods are clearer described and the study appears to be more focussed. However, from the response of the authors to my comments and the unchanged results, I conclude that it was not sufficiently clear what I criticised. The authors arrived at their very high area loss rates for the 2000-2023 period because they changed the definition of a glacier, largely by introducing the vaguely described term 'active ice'. For this, they use (qualitative) geomorphological characteristics of advancing glaciers (L141/2) to define where a glacier terminus is rather than (quantitative) information about glacier flow. As the listed characteristics do not apply to retreating or down-wasting glaciers, they looked at the wrong place for the terminus. How they identified it in the field (when being in the middle of a debris-covered glacier) remains a bit mysterious. They write about repeat photography and that they took ice margins (mapped from multiple Sentinel-2 images) to the field (L122-126), but they neither show the outlines derived from this mapping nor describe how they have determined the region of stagnation without flow fields. Anyway, also stagnant ice has to be included.

The authors state their approach of glacier extent mapping is consistent with prior methods (L117), but it is not. As the authors did not provide glacier outlines for 2023 or 2004, I had to use the 2015/16 outlines from Fountain et al. (2023) for a comparison (Figure 1). This revealed that the authors removed the polygons marked as 'buried ice' by Fountain et al. (2023) for Coe and Eliot glacier although these regions were also included in the study by Lillquist and Walker (2006). This decision (to not say manipulation) resulted in a strong area loss for Coe and Eliot. Also images available in the internet* show the terminus of Eliot at a different place (close to its 2004 position). For Coe one can clearly see the real terminus (marked by a darker frontal part) on the image in Fig. 1B (a bit below the 2004 outline).

For the massive shrinkage of Ladd Glacier the situation is more difficult, as the authors largely followed the interpretation by Fountain et al. (2023), who decided to ignore the debris-covered parts and divided Ladd into five pieces. In fact, these pieces are all still connected and its 2023 extent is not very different from 2004. This might be difficult to see in other images, but in my previous comments I have asked the authors to use the very high-resolution image from the ESRI Basemap for interpretation. This also would have helped to get more realistic extents for Sandy Glacier. For its northern part and terminus region the 2023 extent is still very similar to 2015/16, so their shrinkage here is overestimated. The region with strong area loss since 2004 in its south-western part has indeed little ice left, but the ridge still has a few small glaciers. These have to be included for a correct calculation of area changes. This also applies to Zigzag, where the ice patch marked with a 2 in Fig. 3A and B has to be included; that the ice is probably stagnant does not matter when calculating area changes.

Hence, the strong area loss and retreat of several glaciers reported by the authors for the past 20 years are strongly overestimated for at least three of the larger glaciers, leading to wrong conclusions. In fact, several of the glaciers (Ladd, Coe, Eliot) have barely changed their extents since 2004 when following the interpretation of earlier studies. In the case the authors would like to correct their datasets and report the real area/length changes (please note that providing glacier outlines for review is mandatory), I have a few more comments below. In its current form the results are misleading and should better not be published.

* http://www.mounthoodnationalpark.org/MHNPArticles/210731_Eliot_Branch_Erosion.jpg

[Figure]

*Figure 1: Overlay of glacier outlines (yellow) from Fountain et al. (2023) on a WolrdView 3 image from Sep 2023 available from the ESRI Basemap (World Imagery layer). White arrows point to the polygons that were removed by the authors although the (debris-covered) ice is connected to the main glacier and the real termini are well visible. The cyan circles to the left for Ladd Glacier indicate regions that have been removed from the glacier by Fountain et al. (2023) although the ice is still there and connects the five pieces. This new interpretation results in a strong but wrong area loss from 2004 to 2015/16.*

*Public interest in glacier changes*

As these days any reported glacier changes are of high interest for a large public and critically assessed by 'sceptics', the results presented must be as solid, reproducible and correct as possible. This is not the case here and the wrong assignment of glacier boundaries result in a rather dramatic shrinkage for several glaciers. I would thus not publish the study as it is now and recommend re-working the outlines using the available very high-resolution satellite image. Ladd and Newton Clark Glacier will grow a bit compared to 2015/16, but Zigzag, Eliot and Coe will become comparable to this and earlier extents, allowing calculating consistent change rates from 2004 to 2023. As a note, the authors refer to the year 2004 in Fig. 1B, to 2003 in Table 1 and to 2000 in Table 2. The different dates need to be clarified.

*Field vs. remote sensing*

In my view consistent measurements of glacier length changes is best done in the field, whereas mapping of glacier extent and area changes is the domain of (orthorectified) aerial or satellite imagery acquired under the best possible snow conditions. To obtain meaningful results, the spatial resolution of the imagery used for this should be higher towards smaller glaciers. The 15 m panchromatic band of Landsat is likely the upper resolution limit for the comparably small glaciers of Mt. Hood, mapping with 10 m Sentinel-2 as performed by the authors is certainly more accurate. For this region the authors have also a 0.3 m resolution image with near-perfect snow conditions. Combined with their knowledge about the topography of Mt. Hood and its glaciers, very accurate outlines can be generated based on subtle changes of contrast and/or colour, often also for debris-covered glaciers. I would argue that this is nearly impossible in the field when being on the glacier in a sea of rocks. When including the 'stagnant ice', the 2023 termini for Coe and Eliot will be close to the orange circles in Fig. S1B. As a note, the images displayed in Figures S6 to S8 for Eliot, Coe and Ladd do not show their termini, but regions higher up. So they are of limited use to illustrate changes of the terminus.

*Comparison with other sources*
In Table 1 the authors compare their glacier extents to those from 1981 by Driedger and Kennard (1986) and in Table 2 to year 2000 areas from Jackson and Fountain (2007). But in the latter study the areas are listed for the year 2004 rather than 2000 and for the 1981 extents the authors authors do not show the mapped extents. I assume the authors have these as they provide areas for 1981 with two decimals instead of one. It could then be shown that the interpretation of glacier entities is consistent and related outlines should be added to Fig. 1B. For consistent area changes, all parts belonging to the largest glacier extent need to be included.

*Comments by other reviewers*
In several cases the authors replied that the other reviewers have not mentioned my comment as an issue. Whereas I agree that this is a possibility to ignore a requested change, I think in this case an 'outside' view could still be worth considering as the other reviewers might have worked along the same guidance documents. It is also the right of the authors to say study xyz has been peer-reviewed and published, so the results presented there should be correct, but sometimes this is not the case and different views remain. When it comes to mapping glacier extents this is likely more usual than agreement. In short, I added my comments also because the other reviewers did not notice the issues.

*Relation to climate*
I would strongly recommend removing the climatic interpretation (Figure 6 and Table 3). The sample is too small, each glacier has a different response time, some glaciers did flow over or now end at steep rock cliffs (creating dead ice) and glaciers are debris-covered to a variable extent. All this creates non-linear responses of variable magnitude that should not be compared to some fixed mean values of the climatic history. Of course, glaciers retreat and shrink when it is getting warmer, but that's it. In this region I assume that also precipitation amounts could have an impact on glacier length changes, as these seem to correlate well with small advances of several glaciers. However, due to the backward temporal averaging of the time series this is difficult to say.

Please note that the 920 m retreat of Ladd Glacier between 1984 and 1989 reported by Lillquist and Walker (2006) is also suspicious. The authors of that study have likely decided to interpret the lower part of the glacier in 1989 as dead ice (they write 'ice-cored ground and lateral moraine') and have assigned a new terminus. While this can be required for some glaciers, such jumps in the time series disqualifies it for the statistical analysis performed here (as it relates to a much longer evolution that has little relation to the change in a specific year).

Apart from some further small remarks, I do not further comment on details of the ms, as I expect substantial changes to the text should the authors decide to get the glacier extents corrected and resubmit the study.

**Small comments**
I still found 8 times the word glaciated instead of glacierized.

L398: That all data are available from the figures and tables presented in the text is not true. The key dataset (the glacier outlines from different points in time) where all analysis is based on are not provided. This is mandatory for a proper review of such a detailed study of individual glaciers.

Fig. 3E shows lots of seasonal snow cover. The image should not be used for a visual comparison of glacier extents.

Figs. 4 and 5: The plots need axis bars and tick marks also on the opposite site of the graph.

Fig. 6: Instead of 'Normalized length' (which is hard to imagine) one could also work with length changes. Please also provide the length changes for all glaciers in a table. Now most of them are spread in the glacier descriptions of the supplemental material. Please also use different symbols when using green and red lines in the same plot (maybe replace the green with blue).

---

## Referee Report (RR2)

From the reply of the authors to my earlier comments I conclude that a main point of disagreement is the ***glacier definition***. I start with this point and add details about the climatic interpretation later. As a first step, I had a detailed look at all the publications cited by the authors as well as some additional documents* that I hope would confirm their view to exclude stagnant ice from glacier mapping. Apart from the fact that the authors assign this region subjectively without using flow velocities, I have not found any evidence in the cited and additional literature that buried/stagnant ice had to be removed. If at all, it is mentioned as a possibility rather than mandatory. For example Beason et al. write on P20 (or 1) 'identifying surface velocities on glaciers at the Park to delineate active and stagnant ice'. This means they just distinguish the two types rather than removed the stagnant part and they used flow velocities for it. This is also my interpretation of the polygons marked as 'buried ice' in the inventory by Fountain et al. (2023). These polygons can be marked, but have to be included.

Also Lillquist and Walker (2006) (LW06 below) mention the separation as a possibility, but they are also not applying it. They have seemingly also not applied the rules they have listed for glacier terminus identification. I have compared the dated outlines in their publication to the aerial photography available online** and in particular the oblique 1946 images for Coe and Eliot Glacier reveal strongly down-wasted glacier tongues, basically only consisting of ice-cored lateral moraines. Still, the termini where placed at the end of this assemblage of buried or maybe dead ice remnants. I would thus not take the rules literally.

To conclude this part, I had a look at the UNESCO guidelines by Müller et al. (1977) and the GLIMS Analysis tutorial. The former states: 'Inactive ice must be included in the inventory for hydrological purposes. Marginal and terminal moraines should be included if they contain ice. The "inactive" ice aprons which are frequently to be found above bergschrunds should be regarded as part of the glacier. Glacierets and snow patches of large enough size - if perennial - should also be included in the inventory.' The latter describes under points 4 and 5:
'4. A stagnant ice mass still in contact with a glacier is part of the glacier, even if it supports an old-growth forest.' and '5. All debris-covered parts of the glacier must be included.'

Hence, to be consistent with the earlier interpretations of terminus positions, the literature cited by the authors as well as current guidelines, but also to avoid subjective assignment of glacier areas, the buried or stagnant ice has to be included. I see no way around this. As a way forward, I suggest using the outlines from Fountain et al. (2023), merge the polygons marked as 'buried ice' with the main glacier and then modify them to the 2023 extent, e.g. as visible in the very high resolution satellite image from the ESRI Basemap. Apart from the noticeable retreat of the Coe terminus, also the retreat and lateral melt of Eliot should be updated. For Ladd Glacier one has to first glue the five polygons from Fountain et al. (2023) together, but without the original images a new interpretation of the terminus will be difficult. If the image is not available, I suggest skipping this glacier for this year from the change assessment.

For glaciers that have now two or more parts (e.g. Zigzag/Sandy), all parts included in the former extent have to be summed up for calculation of area changes. I would not do a comparison of change rates when including and excluding the stagnant ice. As you do not have flow velocities, you cannot say where the stagnant ice is and related area or length changes would be arbitrary. So I would skip this stagnant ice discussion altogether. Instead, please have a discussion about the difficulties in identifying termini of down-wasting glaciers. This can be nicely illustrated and allows others to connect to this global-scale mapping problem.

The second main point I want to make is on the ***climatic interpretation***. As mentioned before, most of the glaciers in the study region can hardly be used for a climatic interpretation, as they are either heavily debris covered (e.g. Ladd, Coe, Eliot) or calving (Reid, Newton-Clark). This decouples their responses to climate change and makes calculation of response times challenging. On the other hand, a closer look at the Mt. Hood glacier images available online** combined with the accumulation season precipitation and cumulative length change plots presented by LW06, reveals a fast response of the glaciers to related fluctuations (e.g. advances following increased precipitation). This fast response can likely also be expected from the steepness of the terrain. So, if the authors wish to keep some climatic interpretation, I suggest stripping it down to the basics and continue the time series presented by LW06. This means:

- add a note on the difficulties to interpret glacier fluctuations in this region in climatic terms
- continue the analysis by LW06, showing only curves for winter accumulation and summer ablation without an artificial delay or averaged values
- use cumulative length changes rather than normalized length
- discuss the obviously fast/sensitive reaction to precipitation changes in the past and contrast this with the now dominating retreat/down-wasting due to increasing temperatures (and all the problems related to it for glacier extent mapping).

Regarding the suggested text changes describing the importance of flow for an ice body to be named a glacier, I also suggest to not present it. First, the authors do not use or have flow velocities to apply the definition and second, the cited definition by NSIDC as well as the one by Cogley (2011) says 'evidence of *past* or present movement/flow'. So even if the 'stagnant ice' does not flow today, there is multiple evidence that it has done so in the past. In other words, also these definitions do not allow removing possibly stagnant ice from the area.

I hope the above explanations help clarifying my objections and the authors can revise their ms accordingly. Most of the points in their reply will likely settle when we agree on these main points. As a note, the comments from my earlier reviews are still valid and should thus be considered as well. I have not repeated them here.

* USGS Circular 1132 by Fountain et al. (1997), Mennis and Fountain (2001) in PE&RS and Chapter 17 in the GLIMS Book by Fountain et al. (2014).
** https://glaciers.us/glaciers.research.pdx.edu/image-galleries/oregon-cascades.html

Cogley, J. G.; Hock, R.; Rasmussen, L. A. et al. (2011): Glossary of Glacier Mass Balance and Related Terms; International Hydrological Programme (IHP) of the United Nations Educational, Scientific and Cultural Organization (UNESCO).
Müller, F., Caflisch, T. and Müller, G. (1977): Instructions for Compilation and Assemblage of Data for a World Glacier Inventory. Swiss Federal Institute of Technology Zurich.
Mennis, J.L. and Fountain, A.G. (2001): A spatio-temporal GIS database for monitoring alpine glacier change. Photogrammetric Engineering and Remote Sensing 67 (8), 967–975.

---

## Author Response (AR2)

Editor's and referees' comments in plain text and author responses in *italics*

**Editor (B. Marzeion):**

While the reviewers agree that the revisions lead to a substantial improvement of the manuscript, some rather substantial questions about the methodological decisions taken by the authors remain, and a clarification is likely to have strong impacts on the conclusions of the manuscript (potentially even questioning the "unprecented" included in the title of the manuscript).

*We thank the reviewers for their further insights. However, we note that our methodology is, in fact, well documented and is widely used in the western U.S. To address the questions raised about our methodology, we have now added further discussion of the decision to map actively flowing glacier area rather than include stagnant, debris-covered ice in the mapping. Nevertheless, we would like to clearly point out again that three of the reviewers, including two who work extensively on western U.S. glaciers, had no issues with our mapping of actively flowing glacier area and exclusion of stagnant ice. In fact, one reviewer, M. Pelto, asked us to insert a statement about how robust our methods were.*

*To clarify the impact of our methodological decision on the conclusions of the manuscript, most notably, we now add in the buried ice mapped by Fountain et al. (2023) that they explicitly excluded as part of the glacier (i.e., neither did they list it with the glacier name nor did they include with it its glacier geographical identification number, which they did do for other parts of other glaciers that had turned into snowfields/stagnant ice bodies) to our glacier areas in the discussion section. We also include all other stagnant ice/snowfield bodies that are linked to glaciers by name and glacier geographical identification number. Importantly, these inclusions still find that 21st century retreat is unprecedented with respect to the prior 120 years and that our conclusions do not depend on our methodological choice. Note that we must choose this methodology of mapping and documenting changes in only flowing glacier ice if our results are to be compared to all other studies of western U.S. glaciers and past glacier extents on Mt. Hood. In so doing, we clearly demonstrate that our conclusions are robust regardless of how one defines a glacier.*

**Reviewer 3 (M. Pelto):**

*We appreciate that reviewer 3, M. Pelto, finds our manuscript is significantly improved and does not require further review. We have made his suggested changes in our revision here.*

*Line 38: We have made this change.*

*Line 80: We appreciate this comment.*

*Line 83: We have added this information.*

*Line 125: We have added this sentence and note how this reviewer, M. Pelto, who particularly works on Pacific Northwest glaciers finds our technique to be "robust" and "effective".*

*Line 146: We have reworded this sentence according to the reviewer's suggestions.*

*Line 196: We have reworded this sentence according to the reviewer's suggestions.*

*Line 297: We have removed the word "inferred" as the reviewer asks.*

*Line 300: We appreciate that the reviewer finds our images in Figure 3 to be useful.*

*Line 335: We have added a statement on precipitation's lack of a long-term trend.*

*Line 350: To answer the reviewer's question, no, Frans et al. (2016) did not include any albedo changes beyond a snow-albedo aging function that had maximum snow albedo tuned to 0.80 and then an ice albedo tuned to 0.27. Thus, particulate darkening of the glacier surface from dust/fire-sourced black carbon/etc. was not included. Debris cover influence on mass balance was explicitly modeled.*

*Line 375: We now indicate that these are fall Chinook, summer steelhead, and Coho.*

*Line 385: We have reworded this sentence according to the reviewer's suggestions.*

**Reviewer 4 (F. Paul):**

The revised ms has greatly improved compared to the first version. The now annotated photos guide the reader much better through the observations, the methods are clearer described and the study appears to be more focussed. However, from the response of the authors to my comments and the unchanged results, I conclude that it was not sufficiently clear what I criticised. The authors arrived at their very high area loss rates for the 2000-2023 period because they changed the definition of a glacier, largely by introducing the vaguely described term 'active ice'. For this, they use (qualitative) geomorphological characteristics of advancing glaciers (L141/2) to define where a glacier terminus is rather than (quantitative) information about glacier flow. As the listed characteristics do not apply to retreating or down-wasting glaciers, they looked at the wrong place for the terminus. How they identified it in the field (when being in the middle of a debris-covered glacier) remains a bit mysterious. They write about repeat photography and that they took ice margins (mapped from multiple Sentinel-2 images) to the field (L122-126), but they neither show the outlines derived from this mapping nor describe how they have determined the region of stagnation without flow fields. Anyway, also stagnant ice has to be included.

*We respectfully, but strongly, disagree with the reviewer. We understood his earlier comments but disagree with him on what constitutes a glacier, using a very common definition that has been employed for all prior glacier mapping in the western U.S. Therefore, we did not change the definition of a glacier and we clearly referenced our peer-reviewed methodology. We also described how we determined stagnant ice from actively flowing ice again, all using peer-reviewed methods. Furthermore, the reviewer does not provide any peer-reviewed reference justifying why stagnant ice has to be included. In fact, we find no peer-reviewed publication for*

the western U.S. on the changes in glacier extent that included stagnant ice in their calculations. Inventories do exist that note such stagnant ice, but these inventories do not calculate changes in glacier extents and explicitly exclude stagnant ice from their glacier areas as a separate non-glacier ice body. As such, we find these comments to be the reviewer's subjective opinion rather than based on an objective body of scientific research, like upon which we have based our methods and study.

The authors state their approach of glacier extent mapping is consistent with prior methods (L117), but it is not. As the authors did not provide glacier outlines for 2023 or 2004, I had to use the 2015/16 outlines from Fountain et al. (2023) for a comparison (Figure 1). This revealed that the authors removed the polygons marked as 'buried ice' by Fountain et al. (2023) for Coe and Eliot glacier although these regions were also included in the study by Lillquist and Walker (2006). This decision (to not say manipulation) resulted in a strong area loss for Coe and Eliot. Also images available in the internet* show the terminus of Eliot at a different place (close to its 2004 position). For Coe one can clearly see the real terminus (marked by a darker frontal part) on the image in Fig. 1B (a bit below the 2004 outline).

*In fact, the reviewer is misrepresenting the results of Fountain et al. (2023). Furthermore, the reviewer also seems to overlook the fact that 23 years have passed since Lillquist and Walker (2006) mapped the extent of Eliot and Coe glaciers and that these glaciers have, in fact, changed in the first 23 years of the 21st century.*

*First, we did note the stagnant "buried ice" and that it is now almost fully disconnected from the actively flowing Coe and Eliot glaciers by 2023. In 2016, there was greater physical connection, as seen in Fountain et al. (2023), but Fountain et al. (2023) explicitly mapped these ice bodies as separate non-flowing, non-glacier ice bodies.*

*Second (and to reiterate), Fountain et al. (2023) explicitly did not include these two buried ice bodies in the extents of Coe and Eliot. In fact, they did not consider them to be part of any glacier, which, if the reviewer properly represented their results, would be clear. For glaciers like Glisan or Zigzag that were breaking up into multiple stagnating and stagnant ice bodies in 2015/2016, Fountain et al. (2023) labeled each of these separate polygons with the glacier name, even if it was a snowfield (not a flowing glacier). They also supplied the glacier geographical identification number for each polygon even if it was a snowfield. Contrary to this, Fountain et al. (2003) did not label the separate polygons of buried ice with the name of Coe or Eliot, nor did they supply the glacier identification number for these buried bodies. As such, Fountain et al. (2023) clearly did not consider these two buried ice bodies to be parts of Coe or Eliot glaciers. We thus followed their conclusion. Indeed, we clearly showed the actively flowing glacier outlines of Fountain et al. (2023) and our 2023 outlines, with reviewer 2 (A. Fountain) raising no concerns over our use of his definition of what constitutes Coe and Eliot glaciers.*

*Third, Lillquist and Walker (2006) mapped that ice was actively flowing in 2000 for much of the Coe and Eliot termini that Fountain et al. (2023) mapped as separate stagnant buried ice bodies in 2015/2016. We included the flowing-in-2000 parts of these ice bodies in their length records (Lillquist and Walker, 2006) as well as in their 2000 areas (Jackson and Fountain, 2007) from which we calculate the rate of change. By 2004, at least in the case of Eliot, these ice bodies*

*were stagnating. Jackson and Fountain (2007) determined ice flow and stagnation by measuring boulder movement between 2004 and 2005 for the debris covered terminus of Eliot. They then explicitly excluded the stagnate part of the terminus from their 2004 glacier outlines, parts of the glacier that were still flowing in 2000 as documented by Lillquist and Walker (2006) and Jackson and Fountain (2007). Fountain et al. (2023) subsequently determined that the whole debris covered terminus had stagnated by 2016 and thus explicitly excluded it from Eliot Glacier (and did a similar explicit exclusion for Coe). These actively flowing glacier 2004 outlines were also used as modeling targets by Frans et al. (2016). To then compare our calculated rate of retreat to that simulated by Frans et al. (2016), we must only include changes in the area of actively flowing glacier ice as that is what Frans et al. explicitly modeled.*

*In summary, the reviewer is misinterpreting these results that clearly focused on mapping actively flowing glacier ice and explicitly excluded stagnant ice.*

For the massive shrinkage of Ladd Glacier the situation is more difficult, as the authors largely followed the interpretation by Fountain et al. (2023), who decided to ignore the debris-covered parts and divided Ladd into five pieces. In fact, these pieces are all still connected and its 2023 extent is not very different from 2004. This might be difficult to see in other images, but in my previous comments I have asked the authors to use the very high-resolution image from the ESRI Basemap for interpretation. This also would have helped to get more realistic extents for Sandy Glacier. For its northern part and terminus region the 2023 extent is still very similar to 2015/16, so their shrinkage here is overestimated. The region with strong area loss since 2004 in its south-western part has indeed little ice left, but the ridge still has a few small glaciers. These have to be included for a correct calculation of area changes. This also applies to Zigzag, where the ice patch marked with a 2 in Fig. 3A and B has to be included; that the ice is probably stagnant does not matter when calculating area changes.

*All of these "pieces" that the reviewer refers to are stagnant ice bodies as of 2023 and we excluded them from the extent of Ladd's actively flowing area following the definition of a glacier in the western U.S. The same holds true for Sandy Glacier. We did, in fact, use the 2023 ESRI image (see our Figure 1) as the reviewer requested. However, that image did not change our glacier mapping as we had meticulously over multiple years used numerous images along with high-resolution Google Earth images and extensive field testing. The fact that our final mapping of ice limits did not change with the new ESRI image is a testament to our effort and accuracy. The issue here is not our mapping but rather what one considers part of the glacier. We reiterate that we followed the methodology used for all prior such efforts in the western U.S., whereas the reviewer has a different definition of what constitutes a glacier. Please see more discussion of this issue below.*

Hence, the strong area loss and retreat of several glaciers reported by the authors for the past 20 years are strongly overestimated for at least three of the larger glaciers, leading to wrong conclusions. In fact, several of the glaciers (Ladd, Coe, Eliot) have barely changed their extents since 2004 when following the interpretation of earlier studies. In the case the authors would like to correct their datasets and report the real area/length changes (please note that providing glacier outlines for review is mandatory), I have a few more comments below. In its current form the results are misleading and should better not be published.

*The extents of Ladd, Coe and Eliot have significantly changed in the last 20 years, which is easily seen in the aforementioned studies of Fountain et al. (2023) and Jackson and Fountain (2007). We feel that the reviewer's statement is rooted in a misrepresentation of prior work. (Note, as we said in our previous reviewer response, we will provide the glacier outlines as a shapefile in due course and prior to publication.)*

*However, to address the reviewer's comments in a constructive manner, we have now included a discussion of these two different definitions of a glacier. Specifically, we have added the following paragraph to our methods:*

*"In measuring glacier dimensions, it is important to be clear on what is considered a glacier and thus included in the glacier extents. Clarke (1987) noted in his review of glacier research since the 1820s that "the most interesting property of glaciers is that they flow", using the present tense. The U.S. Geological Survey defines a glacier as "a large, perennial accumulation of crystalline ice, snow, rock, sediment, and often water that originates on land and moves down slope under the influence of its own weight and gravity" (U.S. Geological Survey, 2024, https://www.usgs.gov/faqs/what-glacier). The NSIDC, which hosts data for GLIMS, defines a glacier as "an accumulation of ice and snow that slowly flows over land" (NSIDC, 2024b, https://nsidc.org/learn/parts-cryosphere/glaciers/glacier-quick-facts). However, the NSIDC has another definition of a glacier as: "a mass of ice that originates on land, usually having an area larger than one tenth of a square kilometer; many believe that a glacier must show some type of movement; others believe that a glacier can show evidence of past or present movement" (NSIDC, 2024a, https://nsidc.org/learn/cryosphere-glossary/glacier). Therefore, our mapping of only actively flowing glacier ice follows many traditional definitions of what constitutes a glacier, but we recognize that another definition of a glacier exists that includes ice that used to move and is now stagnant. We return to our decision to map only actively flowing glacier ice, rather than include stagnant ice, in the Discussion section."*

*And then in the Discussion section:*

*"We now return to our finding of unprecedented 21st century retreat of glaciers on Mt. Hood based on mapping only actively flowing glacier ice and comparing 2023 areas to equivalent actively flowing glacier areas in 2015/2016, 2000 and earlier years. If we had followed the other interpretation of what constitutes a glacier and included all stagnant ice in the 2023 glacier extents, would retreat in the 21st century still be unprecedented? As a test, we added to the 2015/2016 and 2023 areas of glaciers (Table 1) the area of additional snow/ice bodies noted in the Supplement mapped by Fountain et al. (2023). We conservatively assumed that these snow and ice bodies did not change in area from 2015/2016 to 2023, which was not the case as some of these ice bodies disappeared (see Supplement). With this additional fixed area, the change in glacierized/perennial snow area on Mt. Hood from 2015/2016 to 2023 was >14±1% versus the change in area for actively flowing glaciers of 18±1%. We thus conclude that defining a glacier as a flowing ice body does not greatly change the percentage of glacier area loss in the last seven to eight years.*

*As another test, we looked at the seven glaciers with records back to 1907 (Fig. 4, 5) (Jackson and Fountain, 2007). We added their peripheral 2015/2016 stagnant buried ice/stagnating ice/perennial snowfield area that Fountain et al. (2023) mapped as separate from the contiguous flowing glacier area to our 2023 glacier areas (Table S2). This additional area was 1.085±0.041 km². About 0.718 km² (~66%) of this area was in the two buried ice bodies that were explicitly mapped as separate from and not parts of Coe and Eliot glaciers as of 2016 (Fountain et al., 2023). Another 12% of this area was in stagnating ice/perennial snowfields that had separated from Ladd Glacier as of 2016. We then compared these new larger glacier/ice/snow areas to the 2000 areas for these seven glaciers, noting that this is not an equal comparison. First, we again assumed that the areas of these peripheral snow/ice bodies did not change in the last seven to eight years, which is not the case. Second, the 2000 areas of these glaciers only included contiguous actively flowing glacier ice and explicitly did not include such peripheral snow ice/bodies or contiguous stagnant ice (Jackson and Fountain, 2007). With these assumptions, we found that these seven glaciers lost area at >1.07±0.1 % yr⁻¹ 2000-2023, which is still unprecedented with respect to 20th century average rate (0.31±0.03 % yr⁻¹) and the fastest 20th century rate (0.56±0.02 % yr⁻¹) (Table S2). Similarly, the retreat of Eliot, Coe, and Ladd glaciers remained unparalleled relative to both the average and fastest 20th century rates despite containing 78% of the non-flowing ice area added to 2023 extents (Table S2). White River's rate of area loss was also unequaled with respect to the 20th century average and remained within uncertainty of its fastest 20th century rate (Table S2). Newton-Clark and Sandy glaciers likewise still had unmatched rates of retreat in the 21st century whereas Reid's rate did not change (no additional ice added) (Table S2). Therefore, our finding of unprecedented 21st century retreat of glaciers on Mt. Hood does not depend on how one defines a glacier."*

**Public interest in glacier changes**
As these days any reported glacier changes are of high interest for a large public and critically assessed by 'sceptics', the results presented must be as solid, reproducible and correct as possible. This is not the case here and the wrong assignment of glacier boundaries result in a rather dramatic shrinkage for several glaciers. I would thus not publish the study as it is now and recommend re-working the outlines using the available very high-resolution satellite image. Ladd and Newton Clark Glacier will grow a bit compared to 2015/16, but Zigzag, Eliot and Coe will become comparable to this and earlier extents, allowing calculating consistent change rates from 2004 to 2023. As a note, the authors refer to the year 2004 in Fig. 1B, to 2003 in Table 1 and to 2000 in Table 2. The different dates need to be clarified.

*First, our outlines do not require reworking because they are based on our use of the definition of a glacier as an ice body that flows. Second, we followed the methods used to map the prior extents of these glaciers in 2015/2016, 2000, 1981, etc. We acknowledge that the reviewer is actually asking us to change to use a different definition of a glacier; however, we respectfully disagree that this change is necessary or practical for our study for the reasons we detail above. However, third and most importantly, as we show above, our finding of unprecedented glacier retreat in the 21st century is robust against how one defines a glacier, addressing any concern over how "sceptics" may look at our work.*

*As for the issue of 2004 versus 2000, we had explained this in our original submission, with the sentence cut in the revisions. We have now added this back:*

*"We focused on the change from 2000 to 2023, rather than 2000 to 2004 and then 2004 to 2023, to make the duration over which the most recent intervals of change were measured closer to the earlier temporal spacing."*

*We also use the 2000 area rather than the 2004 area to calculate changes following the comment/suggestion of reviewer 1. Specifically, he/she/they requested that we determine rates of area change (noting that these are for actively flowing glacier ice alone) from 2000 to 2023 and 1907 to 2000, which we did.*

**Field vs. remote sensing**

In my view consistent measurements of glacier length changes is best done in the field, whereas mapping of glacier extent and area changes is the domain of (orthorectified) aerial or satellite imagery acquired under the best possible snow conditions. To obtain meaningful results, the spatial resolution of the imagery used for this should be higher towards smaller glaciers. The 15 m panchromatic band of Landsat is likely the upper resolution limit for the comparably small glaciers of Mt. Hood, mapping with 10 m Sentinel-2 as performed by the authors is certainly more accurate. For this region the authors have also a 0.3 m resolution image with near-perfect snow conditions. Combined with their knowledge about the topography of Mt. Hood and its glaciers, very accurate outlines can be generated based on subtle changes of contrast and/or colour, often also for debris-covered glaciers. I would argue that this is nearly impossible in the field when being on the glacier in a sea of rocks. When including the 'stagnant ice', the 2023 termini for Coe and Eliot will be close to the orange circles in Fig. S1B. As a note, the images displayed in Figures S6 to S8 for Eliot, Coe and Ladd do not show their termini, but regions higher up. So they are of limited use to illustrate changes of the terminus.

*We agree with the reviewer that the time of image acquisition is critical, which is why we chose to use Sentinel-2 imagery to supplement higher resolution Google Earth imagery and then the ESRI imagery. However, the latter higher resolution imagery often come from inopportune times for determining ice margins, particularly Google Earth. The ESRI imagery is better in timing (albeit still before the lowest point of snow coverage in 2023), but as noted above, it did not change our mapped glacier limits because we used our combined field and remote sensing approach. As we noted, we relied on field measurements that can be real-time compared with the most recent satellite imagery over four years and three field seasons.*

*Furthermore, please note that reviewer 3 (M. Pelto) disagrees with the reviewer (F. Paul) by stating: "The combination of satellite imagery delineation of margins with ground truth mapping relative to the images, is a robust effective technique."*

**Comparison with other sources**

In Table 1 the authors compare their glacier extents to those from 1981 by Driedger and Kennard (1986) and in Table 2 to year 2000 areas from Jackson and Fountain (2007). But in the latter study the areas are listed for the year 2004 rather than 2000 and for the 1981 extents the authors authors do not show the mapped extents. I assume the authors have these as they provide areas for 1981 with two decimals instead of one. It could then be shown that the

interpretation of glacier entities is consistent and related outlines should be added to Fig. 1B. For consistent area changes, all parts belonging to the largest glacier extent need to be included.

*First, see above for 2004 versus 2000 as Jackson and Fountain (2007) provided glacier areas for both years. Second, we report the Driedger and Kennard (1986) areas to the same significant figures place that they reported them (up to three significant figures) after converting millions of feet squared to kilometers squared. The maps/outlines are only available in paper format or scanned paper in PDF; they are not digital otherwise we would have added them to Fig. 1B. And once again, Driedger and Kennard (1986) only mapped actively flowing glacier ice for 1981 as it was a U.S. Geological Survey study, who defines a glacier as an ice body that flows under its own weight. We cannot go back and recreate their mapping and geophysical surveys from 1981 to fit the different definition of a glacier to which this reviewer ascribes.*

**Comments by other reviewers**
In several cases the authors replied that the other reviewers have not mentioned my comment as an issue. Whereas I agree that this is a possibility to ignore a requested change, I think in this case an 'outside' view could still be worth considering as the other reviewers might have worked along the same guidance documents. It is also the right of the authors to say study xyz has been peer-reviewed and published, so the results presented there should be correct, but sometimes this is not the case and different views remain. When it comes to mapping glacier extents this is likely more usual than agreement. In short, I added my comments also because the other reviewers did not notice the issues.

*We respectfully, yet strongly, disagree with the reviewer here. Two of the other reviewers, A. Fountain and M. Pelto, are the leading experts in western U.S. glacier change. We used the same methods for mapping glacier change that they used, including detailed exclusion of stagnant ice from actively flowing ice and only calculating changes in glacier area based on actively flowing ice area. Furthermore, their mapping in the past is based on such qualitative methods for detecting ice flow that we use here. In only one instance has a flow field been produced for a glacier in Oregon, and that is for Eliot Glacier on Mt. Hood. Here, Jackson & Fountain mapped displacement of boulders on the terminus of Eliot. They then used these observations to explicitly exclude from the 2004 Eliot ice margin a still connected yet stagnant section that was flowing in 2000.*

**Relation to climate**
I would strongly recommend removing the climatic interpretation (Figure 6 and Table 3). The sample is too small, each glacier has a different response time, some glaciers did flow over or now end at steep rock cliffs (creating dead ice) and glaciers are debris-covered to a variable extent. All this creates non-linear responses of variable magnitude that should not be compared to some fixed mean values of the climatic history. Of course, glaciers retreat and shrink when it is getting warmer, but that's it. In this region I assume that also precipitation amounts could have an impact on glacier length changes, as these seem to correlate well with small advances of several glaciers. However, due to the backward temporal averaging of the time series this is difficult to say.

*We respectfully disagree with this reviewer that this section should be removed, and our opinion is shared by three other expert reviewers. In fact, reviewer 1 requested we expand (not remove) on this relationship, which we pushed back upon doing because we felt it involved some poorly constrained assumptions. Reviewers 2 (A. Fountain) and 3 (M. Pelto) only requested we address glacier response time in terms of our correlations, which we have done. M. Pelto then found our paper was "significantly improved" and suggested acceptance with some minor suggestions with no further review.*

*Furthermore, while the number of glaciers with length records is only five, we disagree that this is a major issue because all five have significant correlations with temperature, which is consistent with a robust signal. We also feel it is important to assess this glacier-climate relationship as the prior study for this region found that these glaciers were not responding to climate change (Lillquist and Walker, 2006), which was further put forward in Menounos et al. (2019) that concluded there was no change in Mt. Hood glacier mass in the 21st century. As such, this is an important analysis, particularly given the above worries by the reviewer (F. Paul) about climate sceptics.*

*See below response about concerns over glaciers flowing over cliffs and other contributors to non-linear retreat.*

Please note that the 920 m retreat of Ladd Glacier between 1984 and 1989 reported by Lillquist and Walker (2006) is also suspicious. The authors of that study have likely decided to interpret the lower part of the glacier in 1989 as dead ice (they write 'ice-cored ground and lateral moraine') and have assigned a new terminus. While this can be required for some glaciers, such jumps in the time series disqualifies it for the statistical analysis performed here (as it relates to a much longer evolution that has little relation to the change in a specific year).

*We disagree that such changes disqualify Ladd Glacier from such analysis. Furthermore, we feel strongly that such a priori selection of glaciers for climate-length relationships would not be consistent with scientific best practice. However, to constructively address the reviewer's comment, we now have added a sensitivity test to our results section to address this abrupt retreat in Ladd Glacier. We also conduct a similar sensitivity test for White River Glacier, which also has a clearly non-climate related forcing: the development of fumaroles in the upper reaches of White River Glacier in the late 1800s.*

*Specifically in the Results:*

*"Lillquist and Walker (2006) found that the ~560 m retreat of White River Glacier from 1901 to 1938 was facilitated by the development of fumaroles whereas the ~920 m retreat of Ladd Glacier from 1984 to 1989 was due to recession over a cliff (Table S1). To test the significance of these non-climate related recessions on the correlation of these two glaciers' lengths with the records of three different temperature seasons over two backward-running means, we removed these intervals of retreat from the length records and recalculated correlations. White River Glacier's retreat was still significantly correlated (p < 0.05) with 30-year and 15-year backward-running mean-annual, May-October and June-September temperature changes and their r²s changed only at the third decimal place. Ladd Glacier's retreat was still significantly*

*correlated (p < 0.05) with the six temperature-change records, albeit r²s (0.56-0.83) were generally lower than for the full-length records (0.65-0.82).”*

Apart from some further small remarks, I do not further comment on details of the ms, as I expect substantial changes to the text should the authors decide to get the glacier extents corrected and resubmit the study.

*Our glacier extents are correct according to one of the two definitions of a glacier. We clearly state our methods and the definition we used that is widely used in the western U.S. In fact, we know of no study that included stagnant ice in their glacier areas. The one example that the reviewer puts forward is from Fountain et al. (2023); however, as discussed in detail in our response above, the reviewer mischaracterizes the mapping of Fountain et al. (2023) and ignores their explicit exclusion of stagnant ice from the actively flowing glacier area.*

**Small comments**
I still found 8 times the word glaciated instead of glacierized.

*We have made these changes but note that this is also a controversial term with a debated definition going back a century.*

L398: That all data are available from the figures and tables presented in the text is not true. The key dataset (the glacier outlines from different points in time) where all analysis is based on are not provided. This is mandatory for a proper review of such a detailed study of individual glaciers.

*We added the appendix as a Supplement, but the Copernicus submission system only allows one file to be uploaded in this section. So, we think the shape files probably counts as a "Data set", which the system states "Data sets, movies, animations,␣or computer programme code should be deposited in FAIR-aligned data repositories." Alternatively, we could specify the shape files as an "Asset" once we have a DOI in a data repository for it. Given the limitations of the submission system, we made the decision to resubmit everything but the shape files for the revision, and then plan to add the shape file data set/asset prior to publication if the manuscript was accepted for publication (the shape files were not needed for the review process in our opinion but should be available publicly at the time of publication). To reiterate, we will make the shape files available on a FAIR-aligned data repository and provide the DOI as an asset associated with the manuscript. We note that this is the same process that the reviewer (F. Paul) is using for a paper in review at The Cryosphere ("Reconstructed glacier area and volume changes in the European Alps since the Little Ice Age" available at [https://doi.org/10.5194/egusphere-2024-989](https://doi.org/10.5194/egusphere-2024-989)). However, to clarify our intention and address the reviewer's comment, we have now added placeholder text in the Data Availability section of the manuscript (the actual DOI for the asset will be added prior to publication).*

Fig. 3E shows lots of seasonal snow cover. The image should not be used for a visual comparison of glacier extents.

*We respectfully disagree because we feel that the changes we note are clear. For example, we note that reviewer 3 (M. Pelto), who works in the field on glaciers in this part of the world, agrees with us, found our images to be "quite useful", and does not share the opinion of this reviewer (F. Paul).*

Figs. 4 and 5: The plots need axis bars and tick marks also on the opposite site of the graph.

*We have added these to the graphs.*

Fig. 6: Instead of 'Normalized length' (which is hard to imagine) one could also work with length changes. Please also provide the length changes for all glaciers in a table. Now most of them are spread in the glacier descriptions of the supplemental material. Please also use different symbols when using green and red lines in the same plot (maybe replace the green with blue).

*We originally used the change in length for regression. Then, however, reviewer 2 (A. Fountain) asked that we use normalized length instead, which we do in this revised manuscript. We describe normalization in our methods and its application here is reasonable. Namely, this normalization allows to place the change in length of a glacier in perspective of the total glacier length. It is essentially the same process as looking at percent change in glacier area rather than total area change, which this reviewer (F. Paul) asked us to do. As such, we prefer to use the normalized length, consistent with reviewer 2's (A. Fountain's) suggestion.*

*Also, as requested by this reviewer, we now provide in Table S1 the length records for the five glaciers, updating the existing records to 2023.*

*Finally, our shade of green passed the review of one of our authors who is color blind, but we changed the color to blue to alleviate any concerns and address the reviewer's comment.*

---

## Author Response (AR3)

Reviewer 3 comments in plain black text, author replies in *italic black text*

*We appreciate the further suggestions by Reviewer 3 (F. Paul). We have incorporated all suggested changes and note where we made slightly different modifications from what the reviewer suggests.*

From the reply of the authors to my earlier comments I conclude that a main point of disagreement is the **glacier definition**. I start with this point and add details about the climatic interpretation later. As a first step, I had a detailed look at all the publications cited by the authors as well as some additional documents* that I hope would confirm their view to exclude stagnant ice from glacier mapping. Apart from the fact that the authors assign this region subjectively without using flow velocities, I have not found any evidence in the cited and additional literature that buried/stagnant ice had to be removed. If at all, it is mentioned as a possibility rather than mandatory. For example Beason et al. write on P20 (or 1) 'identifying surface velocities on glaciers at the Park to delineate active and stagnant ice'. This means they just distinguish the two types rather than removed the stagnant part and they used flow velocities for it. This is also my interpretation of the polygons marked as 'buried ice' in the inventory by Fountain et al. (2023). These polygons can be marked, but have to be included.

Also Lillquist and Walker (2006) (LW06 below) mention the separation as a possibility, but they are also not applying it. They have seemingly also not applied the rules they have listed for glacier terminus identification. I have compared the dated outlines in their publication to the aerial photography available online** and in particular the oblique 1946 images for Coe and Eliot Glacier reveal strongly down-wasted glacier tongues, basically only consisting of ice-cored lateral moraines. Still, the termini where placed at the end of this assemblage of buried or maybe dead ice remnants. I would thus not take the rules literally.

To conclude this part, I had a look at the UNESCO guidelines by Müller et al. (1977) and the GLIMS Analysis tutorial. The former states: 'Inactive ice must be included in the inventory for hydrological purposes. Marginal and terminal moraines should be included if they contain ice. The "inactive" ice aprons which are frequently to be found above bergschrunds should be regarded as part of the glacier. Glacierets and snow patches of large enough size - if perennial - should also be included in the inventory.' The latter describes under points 4 and 5:
'4. A stagnant ice mass still in contact with a glacier is part of the glacier, even if it supports an old-growth forest.' and '5. All debris-covered parts of the glacier must be included.'

Hence, to be consistent with the earlier interpretations of terminus positions, the literature cited by the authors as well as current guidelines, but also to avoid subjective assignment of glacier areas, the buried or stagnant ice has to be included. I see no way around this. As a way forward, I suggest using the outlines from Fountain et al. (2023), merge the polygons marked as 'buried ice' with the main glacier and then modify them to the 2023 extent, e.g. as visible in the very high resolution satellite image from the ESRI Basemap. Apart from the noticeable retreat of the Coe terminus, also the retreat and lateral melt of Eliot should be updated. For Ladd Glacier one has to first glue the five polygons from Fountain et al. (2023) together, but without the original images

a new interpretation of the terminus will be difficult. If the image is not available, I suggest skipping this glacier for this year from the change assessment.

*We have made all of these changes and adjusted the manuscript accordingly. However, we note these changes do not alter our overall conclusions.*

For glaciers that have now two or more parts (e.g. Zigzag/Sandy), all parts included in the former extent have to be summed up for calculation of area changes. I would not do a comparison of change rates when including and excluding the stagnant ice. As you do not have flow velocities, you cannot say where the stagnant ice is and related area or length changes would be arbitrary. So I would skip this stagnant ice discussion altogether. Instead, please have a discussion about the difficulties in identifying termini of down-wasting glaciers. This can be nicely illustrated and allows others to connect to this global-scale mapping problem.

*We have removed discussion of stagnant ice in glacier area and length changes (removed length changes all together). We have added in a discussion of difficulties in identifying terminus area, focusing on Ladd Glacier with a new Figure 6 of the terminus in 2023 added.*

*However, we do retain the lowest elevation at which we identified flowing ice, which Paul does not raise questions about. In so doing, we just note the change in the lowest elevation of flowing ice found in 2003 versus 2023 following the input of Reviewer 1.*

The second main point I want to make is on the ***climatic interpretation***. As mentioned before, most of the glaciers in the study region can hardly be used for a climatic interpretation, as they are either heavily debris covered (e.g. Ladd, Coe, Eliot) or calving (Reid, Newton- Clark). This decouples their responses to climate change and makes calculation of response times challenging. On the other hand, a closer look at the Mt. Hood glacier images available online** combined with the accumulation season precipitation and cumulative length change plots presented by LW06, reveals a fast response of the glaciers to related fluctuations (e.g. advances following increased precipitation). This fast response can likely also be expected from the steepness of the terrain. So, if the authors wish to keep some climatic interpretation, I suggest stripping it down to the basics and continue the time series presented by LW06. This means:
-add a note on the difficulties to interpret glacier fluctuations in this region in climatic terms
-continue the analysis by LW06, showing only curves for winter accumulation and summer ablation without an artificial delay or averaged values
-use cumulative length changes rather than normalized length
-discuss the obviously fast/sensitive reaction to precipitation changes in the past and contrast this with the now dominating retreat/down-wasting due to increasing temperatures (and all the problems related to it for glacier extent mapping).

*We have greatly reduced our discussion of glacier-climate linkages as the reviewer suggests. In our original manuscript, we already noted the prior issues with glacier fluctuations and climate change in our introduction. In our further revised manuscript, we now just compare the rate of glacier area change (total area, per above reviewer comments) and the change in temperature and precipitation. In so doing, we focus only on the unprecedented 21st century rate of retreat and its correspondence with unprecedented warmth, whereas precipitation is non-unique. We*

*chose to use this approach because it 1) included the records of more glaciers (7 for the period of 1907-2023 and 10 for 1981-2023) and 2) partly avoids issues of down wasting versus length change as contraction of the lateral margins are included, which can occur as the ice thins. We also chose not to continue the length records of Lillquist and Walker (2006) due to the issues identified above in their mapping terminus location. Mainly, Lillquist and Walker (2006) set out to map actively flowing ice but then did not do so, or only partly did so (e.g., for Eliot Glacier they explicitly note in their Figure 3A separation between the 2000 glacier terminus and the lowest extent of debris covered ice in 2000 that would be included in the glacier length/area following Müller et al. (1977)), for the interval where they used only archived photographs according to Review 3's (F. Paul's) investigation. Given this ambiguity, we feel using the glacier area records, which also then includes more glaciers that partly addresses issues of varying climate sensitivity, is a more robust approach. We believe this satisfies the reviewer's concerns even further than the suggested revisions provide above.*

*We have removed the mean annual temperature and total annual precipitation and present the final records with a simple 11-year box-car smoothing that is based on the fastest response time for such glaciers of 11 years (Pelto, 2016). We include this smoothing as our goal is to show how the rate of glacier retreat has changed in accord with climate change, which is an average over a given number of years versus any given year's temperature or precipitation. In the Pacific Northwest, temperature and precipitation have significant year-to-year variability in response to eastern tropical Pacific sea surface temperature fluctuations (i.e., ENSO), which we remove with the 11-year box-car smoothing.*

*We have added a discussion on how past changes in the rate of glacier area change relate to changes in summer temperature and winter precipitation. This includes the overall period of glacier stability and individual glacier advance in the mid 1900s.*

Regarding the suggested text changes describing the importance of flow for an ice body to be named a glacier, I also suggest to not present it. First, the authors do not use or have flow velocities to apply the definition and second, the cited definition by NSIDC as well as the one by Cogley (2011) says 'evidence of *past* or present movement/flow'. So even if the 'stagnant ice' does not flow today, there is multiple evidence that it has done so in the past. In other words, also these definitions do not allow removing possibly stagnant ice from the area.

*We have removed this section.*

I hope the above explanations help clarifying my objections and the authors can revise their ms accordingly. Most of the points in their reply will likely settle when we agree on these main points. As a note, the comments from my earlier reviews are still valid and should thus be considered as well. I have not repeated them here.

*We have made these changes (e.g., only include the photos of Ladd that have slight snow cover in 2003 in the supplement).*